# CODEROSETTA: Pushing the Boundaries of Unsupervised Code Translation for Parallel Programming

**Ali TehraniJamsaz, Arijit Bhattacharjee, Le Chen, Nesreen K. Ahmed**◇
**Amir Yazdanbakhsh**♠, **Ali Jannesari**
*Iowa State University, Ames, Iowa, USA*
`{tehrani, arbhatt9, lechen, jannesari}@iastate.edu`
◇*Cisco Outshift, San Jose, CA, USA*
`nesahmed@cisco.com`
♠*Google DeepMind, Mountain View, CA, USA*
`ayazdan@google.com`

## Abstract

Recent advancements in Large Language Models (LLMs) have renewed interest in automatic programming language translation. Encoder-decoder transformer models, in particular, have shown promise in translating between different programming languages. However, translating between a language and its high-performance computing (HPC) extensions remains underexplored due to challenges such as complex parallel semantics. In this paper, we introduce CODEROSETTA, an encoder-decoder transformer model designed specifically for translating between programming languages and their HPC extensions. CODEROSETTA is evaluated on C++ ↔ CUDA and Fortran ↔ C++ translation tasks. It uses a customized learning framework with tailored pretraining and training objectives to effectively capture both code semantics and parallel structural nuances, enabling bidirectional translation. Our results show that CODEROSETTA outperforms state-of-the-art baselines in C++ to CUDA translation by 2.9 BLEU and 1.72 CodeBLEU points while improving compilation accuracy by 6.05%. Compared to general closed-source LLMs, our method improves C++ to CUDA translation by 22.08 BLEU and 14.39 CodeBLEU, with 2.75% higher compilation accuracy. Finally, CODEROSETTA exhibits proficiency in Fortran to parallel C++ translation, marking it, to our knowledge, as the first encoder-decoder model for this complex task, improving CodeBLEU by at least 4.63 points compared to closed-source and open-code LLMs.[1]

## 1 Introduction

Automatic code translation between programming languages offers numerous benefits, such as modernizing legacy systems, enabling cross-platform development, and refactoring sequential code into parallel high-performance versions. However, this task poses significant challenges, primarily due to the scarcity of parallel corpora—paired datasets of the same applications written in different languages (e.g., C++ ↔ CUDA or Fortran ↔ C++). This lack of data limits the effectiveness of supervised learning approaches. While recent advances in code LLMs have shown promise in general code translation, translating code that involves parallel programming paradigms (e.g., C++ to CUDA) remains largely unexplored. That is primarily due to the inherent complexities in capturing and correctly replicating parallel code semantics [28].

TransCoder [36] and its follow-up works [37, 39] have demonstrated the potential of unsupervised learning for code translation. However, these methods often struggle with the complexities of

---

[1]Code: `https://coderosetta.com`

38th Conference on Neural Information Processing Systems (NeurIPS 2024).

translating between a language and its specialized extensions, such as C++ to CUDA. To address this, BabelTower [46] proposes a CUDA-specific metric and ranking model. Yet, its reliance on language- or library-specific metrics limits its scope, restricting it to unidirectional code translation (C++ → CUDA). Moreover, extending BabelTower to other programming paradigms requires redefining syntax-specific metrics, a process that is both time-consuming and dependent on domain expertise.

To address these limitations, we introduce CODEROSETTA, an encoder-decoder transformer model specifically designed for unsupervised translation between programming languages and their high-performance computing (HPC) parallel extensions. Unlike prior methods that rely on language-specific metrics, CODEROSETTA employs new pre-training and training objectives—including Abstract Syntax Tree (AST) Entity Recognition and customized noise injection strategies for Denoising Auto-Encoding—to learn the inherent features and semantics of code in an unsupervised manner, without relying on language-specific metrics. In summary, this paper makes the following contributions:

- **Unsupervised code translation for parallel programming.** We present CODEROSETTA, an encoder-decoder transformer model tailored for translating between programming languages and their parallel programming extension, specifically targeting C++ to CUDA and Fortran to C++.

- **Customized pre-training and training objectives for code translation to parallel programs.** We introduce two new learning objectives for learning parallel programming syntax and nuances: (1) Abstract Syntax Tree (AST) entity recognition, enabling the model to reason about code structure by identifying and categorizing different syntactic elements, and (2) tailored denoising auto-encoding, incorporating weighted token dropping and insertion, along with an adaptive corruption rate, to help the model discern subtle differences between language constructs and their extensions.

- **Bidirectional translation without language-specific metrics.** Unlike prior works that rely on program-specific metrics for parallel code translation, which narrow the scope of code translation, CODEROSETTA learns bidirectionally (e.g., C++ ↔ CUDA and CUDA ↔ C++) in an unsupervised manner, broadening its scope to different translation tasks.

Our results show that for C++ to CUDA translation, CODEROSETTA achieves a `2.9` BLEU and `1.72` CodeBLUE improvement over existing methods while also increasing compilation accuracy by `6.05%`. Compared to closed-source LLMs, CODEROSETTA's bidirectional approach exhibits even higher gains, with a `19.84` BLEU and `14.39` CodeBLEU improvement, and `2.75%` higher compilation accuracy. To the best of our knowledge, CODEROSETTA is the first model to demonstrate proficiency in the task of Fortran to C++ translation, surpassing the performance of existing closed-source LLMs and open-code LLMs on standard metrics, with up to `4.63`-point improvement in CodeBLEU.

## 2 Related Works

**Automatic parallelization.** Translating from C to CUDA poses a major challenge. Early efforts in this area primarily involved semi-automatic tools that required significant developer intervention. Noaje et al. [30] implemented an OpenMP C [11] to CUDA translation using the OMPi compiler. Other tools, such as CUDAfy.NET and GPUcc [48], provided annotations to assist the translation process. DawnCC [27] automatically annotates C and C++ code for parallelism, utilizing static analysis to identify opportunities for optimizing execution on multicore and GPU architectures with OpenMP/OpenACC directives. However, much of the responsibility for identifying parallelizable sections and optimizing memory usage remained with the developer. Efforts to translate between C/C++ and Fortran have been more limited. FABLE [15] is one of the few frameworks designed for this, facilitating automatic translation of Fortran to C++ while preserving the original code's semantics through advanced analysis and transformation techniques.

**Neural machine translation.** Tournavitis et al. [42] proposed a framework that combines static analysis with machine learning to identify parallelizable code regions and determine the optimal parallelization scheme. This adaptive approach aims to reduce the overhead of manual parallelization while accommodating different architectures. TransCoder [36] pioneered the use of unsupervised learning techniques to translate code across various high-level languages, including Java, C++, and Python, without the need for parallel corpora. Building on TransCoder's architecture, BabelTower [46] extends its capabilities to perform parallel semantic conversion between C and CUDA.

Denoising Auto-Encoding (DAE) has become a popular technique for training encoder-decoder models, as seen in methods like CodeT5 [45] and PLBART [2]. These models typically use common

noising strategies such as masking and token dropping. One of the key differences in the noising strategies used by CODEROSETTA lies in its language-specific characteristics. Rather than random token dropping, CODEROSETTA employs weighted random dropping, prioritizing language-specific reserved keywords to enhance the model's understanding of the target language's semantics. Another unique strategy is token insertion, which encourages the model to differentiate between valid and invalid tokens. These objectives enable CODEROSETTA to better distinguish between different extensions of the same programming language. In summary, CODEROSETTA is a sequence-to-sequence transformer model that learns in an unsupervised manner to translate between programming languages and parallel programming APIs. Additional related work is presented in Appendix J.

## 3 CODEROSETTA: Unsupervised Code Translation for Parallel Programming

This section presents the design and training methodology of CODEROSETTA, our proposed encoder-decoder transformer model for unsupervised code translation. We begin by outlining the overall architecture, followed by a detailed discussion of the pre-training and training objectives that enable CODEROSETTA to effectively capture the nuances of both general-purpose programming languages and their parallel extensions. We focus on the C++↔CUDA and C++↔Fortran translation tasks.

### 3.1 Cross Language Masked Language Modeling

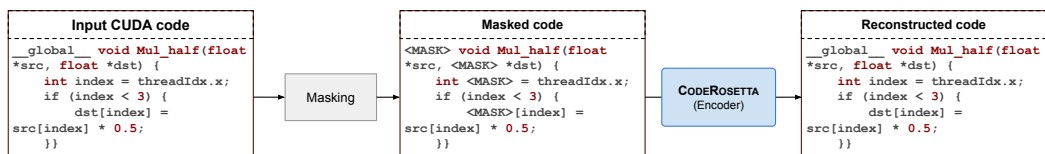

Figure 1: Masked Language Modeling (MLM) pretraining steps in CODEROSETTA.

Pre-training plays a crucial role in enabling transformer models to develop a foundational understanding of programming languages. We use Masked Language Modeling (MLM) [47], a widely adopted pre-training objective, to achieve this, as outlined in Figure 1. In MLM, the model receives input code with a portion of tokens randomly masked. The objective is to predict the masked tokens based on the surrounding context, thereby encouraging the model to learn both local syntactic patterns and broader semantic relationships within code.

To further challenge the model and better reflect code structure, we mask entire words rather than individual tokens. For instance, in the input code snippet " int index", the entire word " index" would be masked, requiring the model to predict the missing identifier based on its type (" int") and its usage in the surrounding code. This approach mirrors how code comprehension often relies on understanding the roles of variables and functions within their scope.

Additionally, while MLM is typically applied to monolingual datasets, we extend it to a cross-lingual setting by training on a combined dataset of both C++ and the target language (CUDA or Fortran). This cross-lingual exposure enables CODEROSETTA to learn shared programming concepts and syntactic structures across languages, such as control flow statements (`if`, `else`, `while`) and variable declarations. By recognizing these commonalities, the model can transfer knowledge across languages, improving its ability to translate even unseen code patterns.

### 3.2 Abstract Syntax Tree Entity Recognition

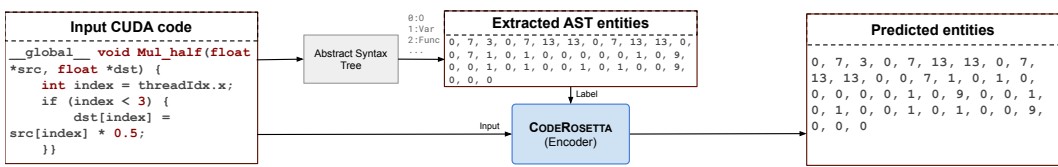

Figure 2: Abstract Syntax Tree Entity Recognition pretraining steps in CODEROSETTA.

Following cross-lingual MLM pre-training, we introduce a new pre-training objective called Abstract Syntax Tree (AST) Entity Recognition (AER) to further improve CODEROSETTA's understanding of code structure. This approach draws inspiration from Named Entity Recognition (NER) in natural language processing [20], where models learn to classify words or phrases into predefined categories (e.g., person, location, or organization). In AER, CODEROSETTA learns to recognize and categorize various syntactic components in code.

The process, illustrated in Figure 2, starts by using Tree-sitter[2], a multi-language parsing library, to generate the Abstract Syntax Tree (AST) of a source code snippet. The AST representation provides a hierarchical, tree-structured view of the code, with each node corresponding to constructs such as *function definitions, variable declarations, or arithmetic expressions*. From this AST, we extract a set of entities and their corresponding categories. Examples of categories used in our implementation include *function*, *variable*, *constant*, *pointer*, and *literal*.

During AER pre-training, CODEROSETTA tokenizes the input code and predicts the syntactic category of each token based on its role in the AST. Tokens that do not correspond to any specific category are labeled as "O" (Outside). This training enables CODEROSETTA to develop an understanding of the syntactic relationships between code elements, an essential step in accurately translating and generating code across different languages and extensions.

A key strength of AER is its flexibility—the set of entity categories can be easily adapted for different languages or programming paradigms. For instance, when focusing on CUDA code, we can introduce specialized categories for parallel constructs such as threadIdx, blockIdx, and gridDim, enabling CODEROSETTA to learn the language-specific semantics of parallel programming.

Furthermore, AER is highly adaptable. Even in cases where AST parsing is only partially available, CODEROSETTA can still leverage this pre-training, showcasing its applicability to diverse code translation tasks. The complete list of tags used in our implementation is provided in Appendix D.2.

### 3.3 Denoising Auto Encoding with Adaptive Noise Injection

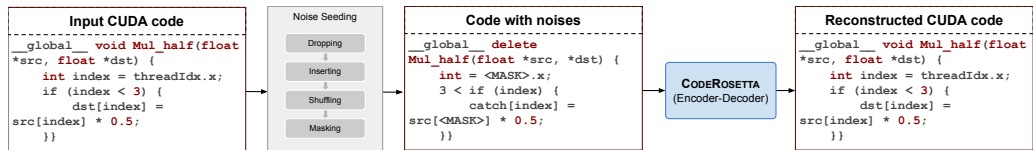

Figure 3: Denoising Auto Encoding.

While cross-lingual MLM and AST Entity Recognition effectively pre-train CODEROSETTA's encoder to generate meaningful representations of source code, the decoder remains untrained at this stage. Consequently, attempting direct code translation would result in suboptimal performance due to the decoder's lack of exposure to the target language's syntax and semantics. To bridge this gap, we employ a Denoising Auto-Encoding (DAE) training strategy specifically tailored for code translation with adaptive noise injection mechanisms. In essence, DAE training involves corrupting the input source code with various types of noise and then training the model to reconstruct the original, noise-free code. This process compels the decoder to learn both the underlying *syntactic rules* of the target language and the ability to recover meaningful code from perturbed inputs, simulating the challenges of translating real-world code with potential variations and inconsistencies.

To initiate the DAE training phase, we first initialize the decoder using the pre-trained encoder's weights, providing it with a starting point for language understanding. Next, we apply a combination of common noise injection techniques, such as random token masking and shuffling, alongside our new noise strategies designed to emphasize the distinctions between programming languages and their extensions. Figure 3 illustrates the overall process of DAE training in CODEROSETTA. We now delve into the specifics of our customized noise injection methods, which distinguish CODEROSETTA from conventional DAE-based code translation models. These strategies are crucial for enabling the model to discern the subtle but significant differences between languages like C++ and their high-performance counterparts like CUDA.

---

[2]https://tree-sitter.github.io

**Weighted token dropping.** To encourage the model to learn the distinctive features of each language and its extensions, we introduce a weighted token dropping strategy during the noise injection phase. Unlike uniform random token removal, this approach assigns higher removal probabilities to language-specific keywords, encouraging the model to focus on critical syntactic elements.

For each programming language or extension, CODEROSETTA maintains a list of reserved keywords. During token dropping, these keywords are prioritized, making them more likely to be removed than other tokens. For example, when training on CUDA code, keywords like `blockIdx`, `threadIdx`, `blockDim`, `__global__`, and `atomicSub` are more frequently targeted for removal.

This weighted sampling creates a more challenging reconstruction task for the model, compelling the decoder to develop a deeper understanding of the language-specific semantics and parallel programming constructs. While the reserved keywords are given higher priority, the weighted random sampling still ensures that other tokens are occasionally dropped, preserving the overall balance of the noise injection process.

**Language-specific token insertion.** In addition to weighted token dropping, we implement a language-specific token insertion strategy to improve CODEROSETTA's ability to discern between languages and their extensions during code generation. This method strengthens the model's robustness against out-of-vocabulary tokens, preventing it from inadvertently blending elements from different languages.

During DAE training, CODEROSETTA must distinguish between valid and invalid tokens within the target language. To facilitate this, we construct a vocabulary of unique tokens for each programming language in our training dataset, tracking their frequency of occurrence. Tokens from the vocabulary of other languages are then randomly inserted into the input code based on their probability from the frequency distribution. For example, in the C++ to CUDA translation task, we insert CUDA-specific tokens into C++ code inputs during DAE training. CODEROSETTA is then trained to recognize and disregard these foreign tokens while reconstructing the original C++ code. This process enables the model to develop an understanding of language boundaries, ensuring it generates syntactically and semantically valid code during translation.

**Adaptive noise ratios** Additionally, we introduce an adaptive noise strategy. Instead of applying a fixed noise ratio, such as 10% for token dropping, we begin with an initial noise ratio and progressively increase it throughout the training process. This approach allows the model to gradually adapt to more challenging conditions as it learns to reconstruct the corrupted input sequences. As the training progresses, the input sequences become increasingly corrupted, making the reconstruction task more difficult and forcing the model to learn more robust representations.

There is a maximum corruption rate that, once reached, halts further increases in noise. This prevents over-corrupting the inputs, ensuring that the model can still derive meaningful patterns. The impact of adaptive noise ratios, along with the new noise strategies, is examined in our ablation study (Section 5.3).

To further support accurate code generation in the target language, we prepend a special `<LANG>` token to each input sequence. During DAE, this token indicates the language of the corrupted input, prompting the decoder to reconstruct the code in the same language. This mechanism ensures that the model remains focused on generating code within the correct language context.

### 3.4 Back Translation for Unsupervised Refinement

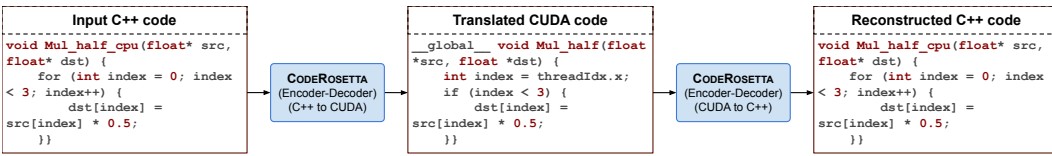

Figure 4: Back Translation.

To further improve CODEROSETTA's translation quality and its ability to capture complex code semantics, we employ back translation during the training process [36]. As illustrated in Figure 4,

this technique leverages the model's bidirectional capability, enabling both source-to-target and target-to-source translations, forming a weakly supervised learning loop.

In back translation, the model is trained on a source-to-target task (e.g., C++ to CUDA) while simultaneously performing the reverse translation (target-to-source, CUDA to C++). For each batch of source code, CODEROSETTA first translates it into the target language. The generated target code is then used as input for a reverse translation, where the model attempts to reconstruct the original source code.

This forward and backward translation cycle provides continuous feedback, allowing CODEROSETTA to compare the reconstructed source code with the original input, thereby learning to detect and correct errors in both translation directions. Through this iterative refinement, the model gradually improves its comprehension of nuanced language differences and complex code structures, resulting in more accurate and semantically consistent translations.

Crucially, we alternate between batches of different language pairs during back translation. This ensures that the model receives balanced exposure to both directions, preventing bias towards a specific language and encouraging the development of robust, generalized translation capabilities.

### 3.5   Finetuning with Synthetic Data from Language Models (Optional Step)

While CODEROSETTA demonstrates promising results through unsupervised training, we explore the potential of further enhancements by leveraging the capabilities of large language models (LLMs) such as GPT-4 [1] and Gemini Ultra [41]. These LLMs, trained on extensive text and code datasets, have exhibited impressive code generation abilities. However, directly employing such large models for code translation can be computationally expensive and impractical for many real-world applications.

To address this, we adopt a knowledge distillation approach [18], where these LLMs serve as teacher models to generate synthetic data for fine-tuning CODEROSETTA, a smaller student model. This method allows us to capture the expertise of the larger models while maintaining computational efficiency.

Specifically, we prompt GPT-4 and Gemini to translate C++ code into CUDA where feasible. After filtering out empty or invalid translations, natural text, and non-relevant data (i.e., instances lacking CUDA-specific keywords), we are left with approximately 5,000 high-quality translations from an initial set of 100,000. This significant reduction highlights the inherent challenges in C++ to CUDA translation.

The resulting synthetic dataset of C++↔CUDA pairs is then used to fine-tune CODEROSETTA. This process allows CODEROSETTA to incorporate the valuable knowledge embedded in the larger LLMs without incurring their high computational costs. It is important to note that this fine-tuning step is *optional* and can be omitted if access to powerful LLMs for synthetic data generation is not feasible.

## 4   Experimental Setup

**Training hyperparameters.** We implement CODEROSETTA using the HuggingFace Transformers library v4.40.1 [47]. The model is a 12-layer encoder-decoder transformer, with each layer having 12 attention heads and a hidden dimension of 1,536. We initialized the tokenizer with a pre-trained Byte Pair Encoding (BPE) tokenizer from UniXcoder [17], which was further trained on our specific training datasets. The training was conducted using the AdamW optimizer [24] and a batch size of 16, using gradient accumulation over two steps. The experiments were run on a single node with four Nvidia A100 SXM4 GPUs, each with 80GB of memory. To speed up the training process, mixed-precision training was enabled. The final model consists of ∼0.8 billion parameters.

### 4.1   Datasets

We evaluate CODEROSETTA on two code translation tasks: C++ to CUDA and Fortran to C++. Table 8 provides an overview of the datasets used. For the C++ to CUDA translation task, we use the dataset from BabelTower [46], which consists of:

- **Unpaired training set:** A collection of 243,008 C++ and CUDA source code files, meeaning there is no direct correspondance between the files in each language. To avoid any language bias, we ensure an equal number of C++ and CUDA files during training.

- **Paired validation and test sets:** The validation set consists of 184 pairs, and the test set has 180 pairs of C++ and CUDA source code files. Each pair represents the same program implemented in both languages, providing a benchmark for evaluating translation accuracy.

For Fortran to C++, no dedicated parallel corpus exists for this specific translation. Thus, we construct our training dataset as follows:

- **Unpaired training set:** We extract the C++ and Fortran subsets from the Stack V2 dataset [25], which includes over 3 billion source code files across more than 600 programming languages. We ensure an equal number of files from each language to prevent bias during training.

- **Paired fine-tuning set:** For fine-tuning, we use the small paired C++-Fortran dataset introduced by Bin et al. [19]. This set is also used for validation.

- **Test set:** To evaluate the final model performance, we use a test set of 33 paired C++ and Fortran programs.

## 4.2 Data Preprocessing

To ensure the quality and consistency of training data, we applied task-specific preprocessing steps for each translation task. **C++ to CUDA.** Although the BabelTower dataset [46] was reportedly cleaned, we found noisy data within the CUDA files. To address this, we curated a list of CUDA-specific reserved keywords and filtered the dataset, retaining only those CUDA files that contained at least one such keyword. This step significantly reduced noise and resulted in a final training set of 243,008 C++ files, matched by an equal number of CUDA files. The validation and test sets remained unchanged, comprising 184 and 180 paired examples, respectively.

**C++ to Fortran.** Preprocessing the Stack V2 dataset for C++ to Fortran translation involved managing the large imbalance between C++ and Fortran files, as well as filtering out low-quality or uninformative code snippets. We implemented the following steps:

- **Educational value filtering:** Inspired by the phi-1 model data filtering approach [16], We randomly sampled 100,000 C++ files from Stack V2 and employed GPT-3.5 to assess their "educational value" for learning C++ coding concepts. We prompted GPT-3.5 (see Figure 5 to classify each snippet as either "Yes" or "No" based on its educational value. These labels were then used to fine-tune a binary classifier built on the CodeSage model [49], which we applied to the remaining C++ files in Stack V2. Only files deemed educationally valuable were retained.

- **Balancing language representation:** From the filtered C++ files, we randomly selected a subset equal in size to the number of Fortran files to create a balanced training set.

- **Length-based filtering:** To ensure training stability and avoid biases toward very short or long code snippets, we filtered out files containing fewer than ten tokens or more than 1,000 tokens in both languages.

After these steps, the final training set for C++ to Fortran translation consisted of 474,856 files. For fine-tuning and validation, we used the small paired C++-Fortran dataset from Bin et al. [19], which contains 282 samples. The model was then evaluated on a test set of 33 paired samples.

```
Determine the educational value of the following code for a student whose goal is to learn C++ coding
↪  concepts. If it has educational value, return only "Yes", else, return "No".
Code:{code}
Educational value:
```

Figure 5: Prompt for determining the quality of C++ source code

## 4.3 Evaluation

To evaluate CODEROSETTA's translations, we use two widely used code translation metrics: BLEU [32] and CodeBLEU [34]. We benchmark CODEROSETTA against the following baselines. For C++ to

Table 1: Summary of C++ to CUDA translation results across various code metrics and compilation accuracy. Second-best results are underlined.

| Model | Static Metrics | | | | Compilation Accuracy (%) |
|---|---|---|---|---|---|
| | BLEU | CodeBLEU | ChrF | ROGUE-L | |
| GPT4 | 46.98 | 64.45 | 70.15 | 63.37 | 96.10 |
| Gemini-Ultra | 57.06 | 61.18 | 73.20 | 69.27 | 80.00 |
| Gemini-Pro | 54.82 | 64.20 | 72.58 | 69.82 | 75.50 |
| DeepSeekCoder | 26.63 | 21.46 | 28.41 | 15.10 | 57.80 |
| StarCoder | 37.58 | 62.58 | 60.16 | 41.84 | 79.40 |
| TransCoder | 72.21 | 71.03 | *N/A* | *N/A* | 83.80 |
| BabelTower | 74.00 | 77.12 | *N/A* | *N/A* | 92.80 |
| CodeRosetta (Ours) | **76.90** | **78.84** | **81.05** | **82.12** | **98.85** |

CUDA, we compare (a) "BabelTower [46]",[3] a state-of-the-art unsupervised code translation model specifically designed for C++ to CUDA translation, and (b) "Transcoder [36]", a general unsupervised code translation model that has demonstrated strong performance on various language pairs. Since a single evaluation metric may capture only one aspect of translation quality [14], we supplement BLEU and CodeBLEU with ROUGE-L [22] and ChrF [33], as recommended by [14]. However, because generated translations from TransCoder and BabelTower were unavailable, ROUGE-L and ChrF scores are only provided for GPT-4, Gemini-Ultra, and Gemini-Pro. We further compare CodeRosetta with two popular open-source code LLMs: StarCoder (starcoder2-7b) [21] and DeepSeekCoder (DeepSeek-Coder-V2-Lite-Base) [12].

For the Fortran to C++ task, we evaluate CodeRosetta against StarCoder [21], an LLM model (15.5B parameters) featuring a decoder-only transformer architecture, fine-tuned on a comprehensive corpus of Fortran code and DeepSeekCoder (DeepSeek-Coder-V2-Lite-Base) [12]. Additionally, we evaluate CodeRosetta alongside several prominent closed-source LLMs, including GPT-4 [1] and Gemini [41], by prompting them to perform code translation using carefully crafted prompts (Appendix I). By evaluating against a broad spectrum of both specialized code translation models and general-purpose LLMs, we effectively gauge CodeRosetta's stranghts and limitations across diverse translation tasks and programming paradigms.

## 5 Experimental Results

### 5.1 C++ to CUDA

Table 1 presents the results of CodeRosetta for C++→CUDA translation. For BabelTower and TransCoder, the results are directly quoted from BabelTower [46], as their models and implementations are not publicly available. Comparing the performance of CodeRosetta to other models, it demonstrates superior translation capabilities for C++ to CUDA. Specifically, CodeRosetta outperforms BabelTower by `2.9` BLEU points. Additionally, it achieves a CodeBLEU score of `78.84`, which is `1.72` points higher than BabelTower. Although GPT4 and Gemini were not specifically trained on this dataset, they still reached CodeBLEU scores of `64.45` and `64.20`, respectively. Evtikhiev et.al [14] indicate that ChrF and ROGUE-L metrics are better suited for code generation tasks than BLEU and CodeBLEU. Notably, CodeRosetta also surpasses these models in both ChrF and ROUGE-L metrics.

CodeRosetta effectively learns the necessary semantics to generate CUDA code without relying on specific metrics for training, a departure from previous approaches. The compilation accuracy of CodeRosetta is `98.85%` after post-processing. For examples of the CUDA code generated by our model compared to other baselines, please refer to Appendix B. Furthermore, CodeRosetta is bidirectional, allowing it to translate both C++ to CUDA and vice versa. Please refer to Appendix A for CUDA to C++ results.

---

[3]We contacted the authors of BabelTower for access to their trained model, source code, and translations but were not able to gain access. Therefore, we cite results directly from their paper.

Table 3: Ablation Study for C++ to CUDA.

| Experiment | Metrics | |
|---|---|---|
| | BLEU ↑ | CodeBLEU ↑ |
| Removing MLM | 52.12 (-24.78) | 51.96 (-26.88) |
| Removing AER | 74.98 (-1.92) | 75.55 (-3.29) |
| Removing DAE (special noises) | 72.41 (-4.49) | 73.22 (-5.62) |
| Removing BT | 75.08 (-1.82) | 73.18 (-5.66) |
| Removing Fine-Tuning | 73.55 (-3.35) | 71.21 (-7.63) |
| Baseline | 76.90 | 78.84 |

### 5.1.1 Post-processing: Compilation Error Analysis

Our test set, consisting of 180 samples, provided diverse input scenarios to evaluate our model's performance. We observed that 23 samples generated compilation errors when processed through the NVCC compiler with the required flags.[4] Upon manual investigation, we found that most errors were trivial and could be easily fixed with minor edits.

Specifically, 48% of the errors were attributed to the use of an undefined generic type T. Another 9% resulted from missing closing braces, while 26% were due to a single missing variable initialization. Additionally, 9% of the errors were caused by incorrect function calls. Only 8% of the files contained no trivial errors. By applying quick fixes for the undefined generic type T, missing variable initializations, and missing closing braces, the overall compilation accuracy significantly improved, with 98.85% of all generated code becoming compilable. This indicates that most errors were simple and could be easily resolved by incorporating compiler feedback, which will be a focus of our future work. Subsection F.1 and Figure 13 in the Appendix presents examples of our findings.

Table 2: Types of compilation errors (28 codes with compilation error out of a total 180 codes).

| Error Type | Percent |
|---|---|
| Undefined generic type T | 48 |
| Missing variable initialization | 26 |
| Missing closing braces | 9 |
| Wrong function call | 9 |
| Non-trivial errors | 8 |

### 5.2 Runtime Evaluation

Although CODEROSETTA demonstrates more accurate translations based on the aforementioned metrics compared to the reference code, these metrics are derived from static evaluations, leaving runtime performance uncertain. To address this, we randomly selected 30 translated CUDA kernels from the test set and created unique template programs to execute them. We ran the translated CUDA kernels using NVCC and found that the functional correctness of the generated code was preserved in the majority of samples (approximately 93%). For further details, see Appendix Section B.

### 5.3 Ablation Study

We conduct an ablation study to evaluate the impact of each training objective on the code translation results of CODEROSETTA. Specifically, we remove individual training objectives (e.g., AER) while keeping the other components intact and retraining the model. Table 3 presents the results of the ablation study for C++ to CUDA translation. As observed, removing any of the pertaining or training objectives negatively impacts translation results, with Masked Language Modeling having the most significant effect when omitted. This is expected, as Masked Language Modeling is the primary pretraining objective that enables the model to understand source code.

**AER training task.** CODEROSETTA employs two pre-training tasks for training its encoder: Mask Language Modeling (MLM) and Abstract Syntax Tree Entity Recognition (AER). In this phase, we maintain consistent training setups except for the removal of the AER component.

---

[4]https://developer.nvidia.com/cuda-11-8-0-download-archive

Table 4: Fortran to C++ translation results.

| Model | CodeBLEU |
|---|---|
| GPT4 | 19.21 |
| Gemini-Ultra | 13.62 |
| Gemini-Pro | 18.91 |
| DeepSeekCoder | 12.09 |
| StarCoder | 18.21 |
| StarCoder (fine-tuned) | 61.30 |
| CODEROSETTA (0.8B) | **65.93** |

**Denoising Auto Encoding.** We also investigate the effectiveness of various noise types and the adaptive corruption rate during Denoising Auto Encoding. For this ablation study, we train the model without weighted token dropping, insertion, and adaptive corruption rate.

**Fine-tuning** Data extraction from larger models is a common practice. In this phase of the ablation study, we evaluate CODEROSETTA's performance without fine-tuning it on the synthetic dataset. From Table 3, we observe that the removal of each proposed learning objective negatively impacts the model's ability to deliver improved code translation.

### 5.4 Fortran to C++

We train and apply CODEROSETTA for translation between C++ and Fortran. Fortran has had a long-standing presence in the scientific computing community; however, its integration with modern HPC systems [38] can pose significant challenges for developers. Due to the complexities involved in translating Fortran to C++, there has been limited effort to address this issue. Bin *et al.* [19] were the first to make significant strides in this area, curating a small paired dataset specifically for this translation task and fine-tuning several open-code LLMs.

They found StarCoder (15B), when fine-tuned, benefited the most from their paired dataset. We compare CODEROSETTA with the fine-tuned StarCoder (15B), as well as with other general LLMs. The results are shown in Table 4. Fine-tuning CODEROSETTA on the dataset from Bin *et al.* [19] further enhances its performance, achieving a CodeBLEU score of 65.93. Notably, CODEROSETTA outperforms StarCoder, even though StarCoder is nearly 20 times larger, highlighting the efficiency of our model. It also surpasses state-of-the-art models like GPT-4 and Gemini by a substantial margin, achieving an improvement of at least 4.63 points in CodeBLEU.

## 6 Conclusion

In this paper, we introduced CODEROSETTA, an encoder-decoder transformer model designed for translating between programming languages and their high-performance computing (HPC) extensions. We proposed two novel learning objectives: Abstract Syntax Tree (AST) Entity Recognition (AER) and customized Denoising Auto-Encoding, which incorporates weighted token dropping and insertion. These contributions enable CODEROSETTA to capture both the general syntactic structure of code and the specific nuances of parallel programming constructs, without relying on language-specific metrics. Our experiments show that CODEROSETTA significantly outperforms state-of-the-art baselines on C++ to CUDA translation, achieving improvements up to 2.9 BLEU, 1.72 in CodeBLEU, and 6.05% in compilation accuracy. Furthermore, CODEROSETTA is, to the best of our knowledge, the first model to demonstrate proficiency in translating Fortran to its parallel counterpart in C++, highlighting its potential in handling diverse programming paradigms.

## Acknowledgment

We would like to thank NSF for their generous support in funding this project (#2211982). In addition, we extend our gratitude to Intel Labs for supporting this project. We also would like to extend our gratitude towards Pengcheng Yin and Chandu Thekkath for their feedback on the early draft of this work. We also appreciate the support from the extended team at Google DeepMind. We thank the Research IT team[5] of Iowa State University for providing access to HPC clusters for conducting the experiments of this research project. We also thank National Center for Supercomputing Applications for providing Delta GPUs through allocation CIS230375 from the Advanced Cyberinfrastructure Coordination Ecosystem: Services & Support (ACCESS) program [7]. Lastly, we would like to express our sincere appreciation to the anonymous reviewers, area chairs, and program chairs of NeurIPS 2024 for their valuable feedback and insights, which significantly contributed to the improvement of this work.

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

# Appendix

## Table of Contents

## A   CUDA to C++ Translation Results

CODEROSETTA is capable of bidirectional translation between languages. Once trained for C++ to CUDA translation, it can also translate CUDA back to C++, unlike previous approaches such as BabelTower [46]. In this section, we compare CODEROSETTA with GPT4 and Gemini on the task of translating CUDA back to C++. Table 5 summarizes the results. As shown, CODEROSETTA demonstrates higher accuracy in translating CUDA to C++. Moreover, we observed that Gemini struggles to clearly distinguish between CUDA and C++, frequently generating C++ translations that are nearly identical to the original CUDA input.

Table 5: CUDA to C++ translation results across different models. We use a similar prompt as the one in Figure 15 with small adjustments.

| Model | BLEU | CodeBLEU |
|---|---|---|
| GPT4 | 70.18 | 68.67 |
| Gemini-Pro | 35.96 | 61.09 |
| CODEROSETTA (Ours) | **77.03** | **71.28** |

# B Functional Correctness Analysis

The metrics and results shown in Table 1 may have limitations in capturing functional equivalence, as discussed by Evtikhiev et al. [14]. To address this, we evaluated the functional correctness of the translated code by compiling and executing the generated programs. For the C++ → CUDA translation task, we randomly selected 30 generated CUDA kernels and developed a template program for their execution. We then compared the runtime results of the translated CUDA code against the reference implementations. Our findings indicate that 93% of the translated CUDA code produced results consistent with the reference.

We analyzed three representative cases of CUDA translation in detail. In the first case, shown in Figure 6, the kernel is designed to be launched with a grid of thread blocks. Each thread calculates its global index `i`, and if `i` is within the array's bounds (`i < N`), it assigns the value `ALPHA` to the element at index `i * INCX` in the array `X`. CODEROSETTA successfully identified the optimal 2D grid structure with `(blockIdx.x + blockIdx.y * gridDim.x) * blockDim.x + threadIdx.x`, whereas other models defaulted to a less efficient 1D structure using `blockIdx.x * blockDim.x + threadIdx.x`. This choice of grid structure significantly impacts CUDA performance, and CODEROSETTA's selection mirrors that of the baseline implementation. Furthermore, CODEROSETTA employed the correct grid structure in four additional instances where other models did not.

The second case, illustrated in Figure 7, involves a kernel designed to initialize an array of offsets for sorting purposes. Each offset corresponds to the starting position of a column in a flattened 2D grid. This is often useful for parallel sorting algorithms or other operations requiring column-wise processing. The expression `int tid = threadIdx.x + blockIdx.x * blockDim.x;` assigns each thread a unique index across the entire grid of blocks, enabling access to distinct elements in a global array. In contrast, the expression `int tid = threadIdx.x;` provides an index that is only unique within a single block. Without proper offset calculations, threads across different blocks could access the same data, potentially leading to race conditions and negating the kernel's intended behavior. This issue was observed in several examples where Gemini-Ultra produced incorrect results due to this oversight.

The third case, depicted in Figure 8, processes 3D arrays in parallel. Each thread calculates its 3D position, checks bounds, and updates specific elements of the array `vec` based on values from `vec1`. The kernel averages and scales values from `vec1`, storing the results in `vec` while ensuring safe memory access within the array's limits. CODEROSETTA correctly handled large block and grid dimensions by using `unsigned long`, whereas both GPT-4 and Gemini-Ultra failed due to the use of `int`, leading to index overflow.

We also analyzed Fortran to C++ translations, shown in Figure 9. The translated code snippets maintained functional equivalence, specifically in the synchronization of shared variables between threads. OpenMP, used in the Fortran code, relies on directives such as `#pragma omp critical`, `#pragma omp flush`, and `#pragma omp atomic` to ensure synchronization and memory visibility. In the C++ translation, equivalent functionality was achieved using `std::mutex`, `std::atomic`, and `std::atomic_thread_fence`. Both approaches ensure that `x` is updated and visible to the second thread before it prints its value, effectively synchronizing the thread actions. CODEROSETTA, Gemini-Pro, and Gemini-Ultra correctly recognized the use of OpenMP in the original code, while GPT-4 did not and opted for a different approach. This highlights the limitations of metrics such as BLEU, which focus on syntax rather than functionality. Despite functional equivalence, GPT-4's translation would score lower due to its syntactic divergence. This underscores the necessity of human evaluation to ensure code correctness, as no single automated metric can fully capture functional behavior.

# C Decontamination Analysis

The C++ to CUDA dataset was obtained from BabelTower [46], which has gone through deduplication and cleaning. Notably, there is no paired trained data available within the dataset, meaning the model does not encounter C++ code alongside the CUDA equivalent during training. As such, the model must rely solely on self-supervised training objectives to learn to embed source code from different

```cuda
__global__ void fill_kernel(int N, float ALPHA,
    float *X, int INCX) {
    int i = (blockIdx.x + blockIdx.y *
        gridDim.x) * blockDim.x + threadIdx.x;
    if (i < N) {
        X[i * INCX] = ALPHA;
    }
}
```

(a) Reference CUDA kernel.

```cuda
__global__ void fill_gpu(int N, float ALPHA,
    float *X, int INCX) {
    int i = (blockIdx.x + blockIdx.y *
        gridDim.x) * blockDim.x + threadIdx.x;
    if (i < N) {
        X[i * INCX] = ALPHA;
    }
}
```

(b) CODEROSETTA generated CUDA.

```cuda
__global__ void fill_gpu(int N, float ALPHA,
    float *X, int INCX) {
    int i = blockIdx.x * blockDim.x +
        threadIdx.x;
    if (i < N) {
        X[i * INCX] = ALPHA;
    }
}
```

(c) GPT4 generated CUDA.

```cuda
__global__ void fill_gpu(int N, float ALPHA,
    float *X, int INCX) {
    int i = blockIdx.x * blockDim.x +
        threadIdx.x;
    if (i < N) {
        X[i * INCX] = ALPHA;
    }
}
```

(d) Gemini Ultra generated CUDA.

Figure 6: CUDA kernel function to fill an array `X` with a constant value `ALPHA`, parallelized across threads.

```cuda
__global__ void set_sorting_offset(const int
    nrows, const int ncols, int *offsets) {
    int tid = threadIdx.x + blockIdx.x *
        blockDim.x;
    if (tid < ncols) {
        offsets[tid] = tid * nrows;
    }
    return;
}
```

(a) Reference CUDA kernel.

```cuda
__global__ void set_sorting_offset(const int
    nrows, const int ncols, int *offsets) {
    int tid = blockIdx.x * blockDim.x +
        threadIdx.x;
    if (tid <= ncols) {
        offsets[tid] = tid * nrows;
    }
    return;
}
```

(b) CODEROSETTA generated CUDA.

```cuda
__global__ void set_sorting_offset_cuda(const
    int nrows, const int ncols, int *offsets) {
    int tid = threadIdx.x + blockIdx.x *
        blockDim.x;
    if (tid <= ncols) {
        offsets[tid] = tid * nrows;
    }
}
```

(c) GPT4 generated CUDA.

```cuda
__global__ void set_sorting_offset(const int
    nrows, const int ncols, int* offsets) {
    int tid = threadIdx.x;
    if (tid < ncols) {
        offsets[tid] = tid * nrows;
    }
}
```

(d) Gemini Ultra generated CUDA.

Figure 7: CUDA kernel to set sorting offsets based on row and column indices, where each thread computes an offset value for a given column index.

languages into a shared embedding space. Paired data is available only in the test set, which we used for evaluating the model's performance.

To assess the potential overlap between the test and the training data from BabelTower, we used CodeBERTScore [51] to measure similarity.

Table 6: C++↦CUDA Decontamination Analysis.

| Data | CodeBERTScore range | | | | | |
|---|---|---|---|---|---|---|
| | 0.4-0.5 | 0.5-0.6 | 0.6-0.7 | 0.7-0.8 | 0.8-0.9 | 0.9-1.0 |
| C++ ↔ CUDA Train Data | 0% | 1.7% | 44.80% | 48.61% | 4.78% | 0.03% |
| C++ ↔ CUDA Synthetic Data | 0% | 0.8% | 33% | 58% | 7% | 0.05% |

Table 6 presents the distribution of CodeBERT scores and the corresponding amount of data in each range. For example, `48.61%` of training data achieved a CodeBERTScore between 0.7 and 0.8 when

```cuda
__global__ void opL23(float *vec, float *vec1,
    long depth, long rows, long cols) {
  unsigned long x = threadIdx.x + blockIdx.x *
      blockDim.x;
  unsigned long y = threadIdx.y + blockIdx.y *
      blockDim.y;
  unsigned long z = threadIdx.z + blockIdx.z *
      blockDim.z;
  unsigned long long i = z * rows * cols + y *
      cols + x;
  unsigned long long j = z * rows * cols + y *
      cols;
  unsigned long size2d = cols;
  unsigned long size3d = depth * rows * cols +
      rows * cols + cols;
  if (x >= cols  y >= rows  z >= depth)
    return;
  if (i + cols + 1 >= size3d) return;
  vec[i + cols] = 0.5 * (vec1[i + cols] +
      vec1[i]);
  if (j + 1 >= size2d) return;
  vec[j] = 0.5 * (vec1[j]);
}
```

(a) Referece CUDA kernel.

```cuda
__global__ void opL23_gpu(float *vec, float
    *vec1, long depth, long rows, long cols) {
  unsigned long x = blockIdx.x * blockDim.x +
      threadIdx.x;
  unsigned long y = blockIdx.y * blockDim.y +
      threadIdx.y;
  unsigned long z = blockIdx.z * blockDim.z +
      threadIdx.z;
  unsigned long long i = z * rows * cols + y *
      cols + x;
  unsigned long long j = z * rows * cols + y *
      cols;
  unsigned long size2d = cols;
  unsigned long size3d = depth * rows * cols +
      rows * cols + cols;
  if (i + cols + 1 >= size3d) return;
  vec[i + cols] = 0.5 * (vec1[i + cols] +
      vec1[i]);
  if (j + 1 >= size2d) return;
  vec[j] = 0.5 * (vec1[j]);
}
```

(b) CODEROSETTA generated CUDA.

```cuda
__global__ void opL23_gpu(float *vec, float
    *vec1, long depth, long rows, long cols) {
  int x = blockIdx.x * blockDim.x +
      threadIdx.x;
  int y = blockIdx.y * blockDim.y +
      threadIdx.y;
  int z = blockIdx.z * blockDim.z +
      threadIdx.z;
  if (x < cols && y < rows && z < depth) {
    unsigned long long i = z * rows * cols +
        y * cols + x;
    unsigned long long j = z * rows * cols +
        y * cols;
    unsigned long size2d = cols;
    unsigned long size3d = depth * rows *
        cols + rows * cols + cols;
    if (i + cols + 1 < size3d) {
      vec[i + cols] = 0.5 * (vec1[i +
          cols] + vec1[i]);
    }
    if (j + 1 < size2d) {
      vec[j] = 0.5 * (vec1[j]);
    }
  }
}
```

(c) GPT4 generated CUDA.

```cuda
__global__ void opL23_gpu(float *vec, float
    *vec1, long depth, long rows, long cols) {
  int x = blockIdx.x * blockDim.x +
      threadIdx.x;
  int y = blockIdx.y * blockDim.y +
      threadIdx.y;
  int z = blockIdx.z * blockDim.z +
      threadIdx.z;
  if (x < cols && y < rows && z < depth) {
    unsigned long long i = z * rows * cols + y *
        cols + x;
    unsigned long long j = z * rows * cols + y *
        cols;
    unsigned long size2d = cols;
    unsigned long size3d = depth * rows * cols +
        rows * cols + cols;
    if (i + cols + 1 < size3d) {
      vec[i + cols] = 0.5 * (vec1[i + cols] +
          vec1[i]);
    }
    if (j + 1 < size2d) {
      vec[j] = 0.5 * (vec1[j]);
    }
  }
}
```

(d) Gemini Ultra generated CUDA.

Figure 8: CUDA kernel opL23, averaging 3D grid elements from vec1 into vec, with boundary checks.

compared against test data. Ranges with no data are omitted. A score below 0.8 indicates low or moderate similarity. As shown, the majority of the training samples exhibit a CodeBERTScore below 0.8, reflecting minimal similarity to the test set. A similar trend was observed when we applied this analysis to the synthetic dataset.

# D   Unsupervised Training Parameters

## D.1   Training Parameters

For Masked Language Modeling (MLM) training, we use a learning rate of $8 \times 10^{-5}$ and train for 100 epochs with 15% masking. After each epoch, we measure the perplexity on the validation set and save the model if the perplexity is the lowest. For Abstract Syntax Tree (AST) entity recognition, we use a learning rate of $5 \times 10^{-6}$ and train for ten epochs. We then create the encoder-decoder

model by transferring the encoder's weights to initialize the decoder, so the decoder begins with some foundational knowledge.

For Denoising Auto-Encoding and Back Translation, we use a learning rate of $5 \times 10^{-5}$ and train for 20 epochs. For Denoising Auto-Encoding, we set the masking to 15%, token dropping to 25%, and token insertion to 15%, with a denoising ratio increasing by 2.5% per epoch. Finally, for fine-tuning, we use a learning rate of $5 \times 10^{-5}$ for ten epochs. At each training iteration, we save the model with the lowest validation loss. All the parameter values are determined empirically through detailed hyperparameter tuning.

### D.2    AST Entity Recognition Tags

Table 7: AER Tags.

| Tag ID | Tag Type |
|--------|----------|
| 1 | identifier/variable |
| 3 | function |
| 5 | type identifier |
| 7 | primitive type (int, float, etc.) |
| 9 | number literal |
| 11 | & pointer expression/reference |
| 13 | * pointer declarator |
| 15 | constant |

The AER tags used in pretraining are shown in Table 7.

### D.3    Dataset Statistics

A detailed overview of the dataset is shown in Table 8.

## E    Impact of Beam Size

We conducted beam search decoding with varying beam sizes, returning the top candidate in each case. The results, shown in Table 9, indicate that CODEROSETTA consistently produces the same output, regardless of the beam size.

## F    Analysis of Generated Code from CODEROSETTA and Closed-Source LLMs

**C++ → CUDA:** In this part, we compare the code generated by CODEROSETTA, GPT4, and Gemini-Ultra. As the BabelTower model and its code are not publicly available, we were unable to access them. However, the BabelTower paper highlights a kernel where the model failed to generate CUDA code due to a syntax error when defining `keyCharPtr`, as shown in Figure 10. In contrast, CODEROSETTA successfully generates the correct CUDA code. It is interesting to note that CODEROSETTA also recognized the `if` condition and improved the readability of the code by inverting the `if` statement, similar to the approach taken by Gemini-Ultra and GPT4. Additionally, CODEROSETTA adheres to the preferred practice of declaring a variable or pointer before assigning a value, which is why first `keyCharPtr` is defined out of the `if` statement.

We demonstrate another example in Figure 11, where CODEROSETTA accurately reproduces the reference CUDA kernel without adding unnecessary lines of code, such as a host or main function, which is often seen in other models.

**Fortran → C++:** Figures 9, 12 show examples of C++ code generated by CODEROSETTA in comparison with other LLMs. Despite CODEROSETTA's smaller size, it effectively translates Fortran code into correct C++ code.

Moreover, we also evaluated our model in terms of C++ → Fortran translation 10. The results indicate the capability of CODEROSETTA in translating to and from Fortran code.

Table 8: Dataset statistics for C++, CUDA, and Fortran programming languages.

| Programming Pair | Train | Valid | Test | Size |
|---|---|---|---|---|
| C++ ↔ CUDA | 243,008 (unpaired) | 184 | 180 | 626.1 MB (Train) |
| | | | | 139.1 KB (Valid) |
| | | | | 141.9 KB (Test) |
| C++ ↔ Fortran | 474,856 (unpaired) | N/A | 33 | 1.2 GB (Train) |
| | 282 (paired) | | | 99.0 KB (Test) |

Table 9: Effect of different beam sizes on C++ to CUDA translation.

| Beam Size | Metrics | |
|---|---|---|
| | BLEU | CodeBLEU |
| 1 | 76.47 | 78.43 |
| 5 | 76.90 | 78.84 |
| 10 | 76.85 | 78.87 |
| 25 | 76.70 | 78.67 |
| 50 | 76.61 | 78.65 |

Table 10: C++ to Fortran translation results in terms of CodeBLEU.

| Model | CodeBLEU |
|---|---|
| GPT4 | 35.32 |
| Gemini-Ultra | 33.64 |
| Gemini-Pro | 32.36 |
| CODEROSETTA (Ours) | **70.46** |

## F.1 Common Issues and Post-processing in CODEROSETTA-Generated Code

Code translated by large language models like GPT-4 often includes additional caller functions that extend beyond the scope of the original function. In contrast, code translated by CODEROSETTA may occasionally fail to compile despite being syntactically correct. We identified two common issues in the code generated by CODEROSETTA and applied a simple post-processing method to ensure a fair comparison across models.

The first issue involves the use of generic types, which can enhance code efficiency but require explicit type definitions at compile time. Figure 13a shows the use of a generic type, although the necessary definition is missing. Adding the type definition, as shown in Figure 13b, resolves the compilation issue. The second issue relates to misses variable initialization in the function definition, as shown in Figure 13c. By initializing the required variable, as demonstrated in Figure 13d, the compilation problem is resolved. Lastly, for longer code snippets, CODEROSETTA occasionally omits the closing curly bracket.

## G  Discussion on Unsupervised Training

### G.1  Fine-tuning for Code Translation

In the context of code translation, paired data is scarce. However, our model benefits from a strong foundational understanding of code translation acquired through unsupervised and self-supervised pre-training on 243K training examples for C++ ↔ CUDA. We demonstrate that fine-tuning, even with a small amount of synthetic data—without verifying the one-to-one mapping between the generated samples and the input code in a supervised manner—can further improve the model's performance. Specifically, fine-tuning with merely 5K paired samples (less than 2% of total data) generated by larger

models still led to significant performance gains. While synthetic data may introduce some errors (as large models can make translation mistakes), the combination of this foundational pre-training and fine-tuning with a small synthetic dataset yields further improvements.

## G.2 Back Translation

Back Translation (BT) has been extensively used in unsupervised translation tasks for both natural language and code. We integrate this technique with the denoising auto-encoding (DAE) objective, ensuring that the model is not trained exclusively on a single objective. During training, the model alternates between DAE and BT for each batch of data. This prevents the model from relying solely on BT and 'cheating' by outputting the input source code as an intermediate translation. To better understand this behavior, we analyzed the intermediate outputs during back translation.

For instance, Figure 14 shows a C++ input and its corresponding intermediate CUDA translation. As shown, while the model attempts to translate the code to CUDA, the output contains errors, such as the undefined variable `j`. In the back translation process, this noisy CUDA code output is fed back into the model, which then attempts to reconstruct the original C++ input. Since the model alternates between languages during back translation, it occasionally generates noisy CUDA or C++ code. This approach improves the model's robustness when handling noisy inputs in translation tasks.

## H    Translation Pitfalls: Invalid Tokens in Target Language

During translation between programming languages (e.g., from C++ to CUDA), certain entities, libraries, and syntaxes present in the source language may not be valid or supported in the target language. For example, C++ Standard Template Libraries (STL) such as `std::unique_ptr` are not compatible with CUDA's device code and must be excluded from translations. The pre-training process in CODEROSETTA equips the model with semantic knowledge of both source and target languages, reducing the frequency of invalid tokens during translation. Nonetheless, there are still instances where the model may fail to correctly map common source language entities to valid target language counterparts.

While our test set contained no occurrences of `std::unique_ptr`, we deliberately included this construct in a separate C++ code example to evaluate CODEROSETTA's handling of STL-specific constructs. Figure 16 demonstrates this case, where the model successfully generates CUDA code by omitting the unsupported `std::unique_ptr` in the device kernel. Instead, the use of `std::unique_ptr` is correctly retained in the host kernel, specifically in the `main` function, which runs on the CPU. Since CODEROSETTA is trained to focus on device function generation, the translation is accurate in this instance.

On the other hand, Figure 17 illustrates a case of incorrect translation, where CODEROSETTA, along with other large closed-source models like GPT-4, Gemini-Ultra, and Gemini-Pro, failed to generate valid CUDA code. The translated code includes the line `*rho = 0;`, which initializes the `rho` variable to zero. In a multi-threaded GPU environment, executing this kernel across multiple threads and blocks simultaneously can lead to a race condition, as multiple threads would attempt to write to the same memory location concurrently. Without synchronization mechanisms like atomic operations or reduction techniques, this results in unpredictable and incorrect behavior. The correct approach would be to initialize `rho` in the host code and use `atomicAdd` to accumulate values in the device code safely.

## I    Prompt Template and LLMs

In this section, we describe the prompt template used to translate between different programming languages and libraries. The template, shown in Figure 15, served as the basis for all translation tasks, with language-specific adjustments made by updating the source and target languages as required. For this study, we use OpenAI API's GPT-4 API, using a fixed temperature of zero to ensure deterministic outputs across all models, including CODEROSETTA. All queries were executed on May 18th, 2024, ensuring consistency in results throughout the experiments.

## J   Additional Related Work

**Automatic parallelization.** Early efforts in auto-parallelization were primarily focused on identifying independent loops that could be executed in parallel. Renowned compilers like the Portland Group (PGI) and Intel's C++ Compiler (ICC) have embedded auto-parallelization capabilities, offering pragma-based hints to guide the parallelization process. These compilers analyze loop dependencies, data flow, and potential side effects to generate parallel code, often targeting OpenMP or MPI for multi-threading and distributed computing, respectively. The advent of Polyhedral model-based tools marked a significant advancement in auto-parallelization techniques. The Polyhedral model [6] offers a powerful algebraic representation for optimizing loop nests with affine bounds and access patterns. Pluto [8] is an auto-parallelization tool that utilizes the Polyhedral model to perform loop transformations, tiling, and fusion for effective parallel execution while considering data locality optimization. PPCG (Polyhedral Parallel Code Generation) [44] is another tool that exploits the polyhedral model to automatically optimize and generate parallel code from high-level abstractions, targeting multicore CPUs and GPUs.

**Neural machine translation.** TransCoder-ST [37] extends the original work [36] by adding automated unit testing. TransCoder-IR [39] extends it even further by exploiting LLVM IR for program translation. HPC-GPT [13] uses GPT4 to create an instruction-answer dataset for two tasks (AI models and datasets for HPC and data race detection), then Llama model [43] is supervised tuned on this dataset. Pan et al. [31] provided one of the first studies on the types of errors that are often produced in code translation.

There is a growing number of large language models (LLMs) for code generation [5, 35, 40, 50, 26, 4, 29, 3]. Most of these works focus mainly on natural language to code generation. Although these Code LLMs can generate code in various programming languages, Python, in particular, has received more attention compared to others. This could be due to the number of available benchmarks that assess Python coding capabilities [10, 23], though other programming languages have been gaining more attention recently as well [9]. Despite the growing number of Code LLMs, these models are typically not specifically trained for code translation, even though they can perform code translation to some extent, as shown by Pan et al. [31].

## K   Limitations

While CODEROSETTA demonstrates promising results in code translation, several avenues for future work exist. Currently, CODEROSETTA targets C++ $\rightarrow$ CUDA and Fortran $\rightarrow$ C++ translation. Extending its capabilities to encompass a wider range of HPC languages and parallel programming models would further broaden its scope. In addition, we plan to improve the set of entity categories used in AER to capture a better representation of code semantics. This will involve incorporating additional tags for constructs like data types, control flow mechanisms, and parallel programming-specific primitives.

```c
#include <stdio.h>
#include <omp.h>

int main() {
    int x = 0, y;
    #pragma omp parallel num_threads(2)
    {
        int thrd = omp_get_thread_num();
        if (thrd == 0) {
            #pragma omp critical
            {
                x = 10;
            }
            #pragma omp flush(x)
            #pragma omp atomic write
            y = 1;
        } else {
            int tmp = 0;
            while (tmp == 0) {
                #pragma omp atomic read acquire
                tmp = y;
            }
            #pragma omp critical
            {
                if (x != 10) printf("x = %d\n",
                    x);
            }
        }
    }
    return 0;
}
```

(a) Reference C++.

```c
#include <stdio.h>
#include <omp.h>

int main() {
    int x = 0, y;
    #pragma omp parallel num_threads(2)
    {
        int thrd = omp_get_thread_num();

        if (thrd == 0) {
            #pragma omp critical
            {
                x = 10;
            }
            #pragma omp flush
            #pragma omp atomic write
            y = 1;
        } else {
            int tmp = 0;
            while (tmp == 0) {
                #pragma omp atomic read acquire
                tmp = y;
            }
            #pragma omp critical
            {
                if (x != 10) {
                    printf("x = %d\n", x);
                }
            }
        }
    }
    return 0;
}
```

(b) CODEROSETTA generated C++.

```cpp
#include <iostream>
#include <atomic>
#include <thread>
#include <mutex>
std::atomic<int> x(0);
std::atomic<int> y(0);
std::mutex mtx;
void threadFunc(int thrd) {
    if (thrd == 0) {
        {
        std::lock_guard<std::mutex> lock(mtx);
        x.store(10, std::memory_order_relaxed);
        }
        std::atomic_thread_fence(
        std::memory_order_release
        );
        y.store(1, std::memory_order_relaxed);
    } else {
        int tmp = 0;
        while (tmp == 0) {
        tmp =
            y.load(std::memory_order_acquire);
        }
        {
        std::lock_guard<std::mutex> lock(mtx);
        std::cout << "x = "
          << x.load(std::memory_order_relaxed)
          << std::endl;
        }
    }
}
int main() {
    std::thread t1(threadFunc, 0);
    std::thread t2(threadFunc, 1);
    t1.join();
    t2.join();
    return 0;
}
```

(c) GPT4 generated C++.

```cpp
#include <iostream>
#include <omp.h>

int main() {
    int x, y, thrd;
    int tmp;

    x = 0;

    #pragma omp parallel num_threads(2)
        private(thrd, tmp)
    {
        thrd = omp_get_thread_num();

        if (thrd == 0) {
            #pragma omp critical
            {
                x = 10;
            }
            #pragma omp flush(x)
            #pragma omp atomic write
            y = 1;
        } else {
            tmp = 0;
            while (tmp == 0) {
                #pragma omp atomic read acquire
                tmp = x;
            }
            #pragma omp critical
            {
                std::cout << "x = " << x <<
                    std::endl;
            }
        }
    }

    return 0;
}
```

(d) Gemini Ultra generated C++.

Figure 9: A C++ OpenMP example with thread sync using atomic operations and critical sections.

```
void kernelXor(unsigned int key, char*
    input_str_cuda, unsigned char*
    possible_plaintext_str_cuda, int
    input_length) {
    int id;
    char* keyCharPtr;

    for(id = 0; id < input_length; id++) {
        int keyIndex = id % 4;
        keyCharPtr = (char*)&key;
        char keyChar = keyCharPtr[keyIndex];
        possible_plaintext_str_cuda[id] =
            keyChar ^ input_str_cuda[id];
    }
}
```

(a) C++ code.

```
__global__ void kernelXor(unsigned int key,
    char *input_str_cuda, unsigned char
    *possible_plaintext_str_cuda, int
    input_length) {
    int id = threadIdx.x + blockDim.x *
        blockIdx.x;
    if (id >= input_length) return;

    int keyIndex = id % 4;
    char *keyCharPtr = ((char *)&key);
    char keyChar = keyCharPtr[keyIndex];

    possible_plaintext_str_cuda[id] = keyChar ^
        input_str_cuda[id];
}
```

(b) CUDA reference.

```
__global__ void kernelXor(
    unsigned int key,
    char *input_str_cuda,
    unsigned char *possible_plaintext_str_cuda,
    int input_length)
{
    int id = blockIdx.x * blockDim.x +
        threadIdx.x;
    if (id >= input_length)
        return;
    int keyIndex = id % 4;
    int keyCharPtr = ((char *)&key);
    char keyChar = keyCharPtr[keyIndex];
    possible_plaintext_str_cuda[id] = keyChar ^
        input_str_cuda[id];
}
```

(c) BabelTower.

```
__global__ void kernelXor(unsigned int key,
    char* input_str_cuda, unsigned char*
    possible_plaintext_str_cuda, int
    input_length) {
    int id = blockIdx.x * blockDim.x +
        threadIdx.x;
    char* keyCharPtr;

    if (id < input_length) {
        int keyIndex = id % 4;
        keyCharPtr = ((char*)&key);
        char keyChar = keyCharPtr[keyIndex];
        possible_plaintext_str_cuda[id] =
            keyChar ^ input_str_cuda[id];
    }
}
```

(d) CODEROSETTA.

```
__global__ void kernelXor(unsigned int key,
    char *input_str_cuda,
        unsigned char
            *possible_plaintext_str_cuda,
        int input_length) {
    int id = blockIdx.x * blockDim.x +
        threadIdx.x;
    if (id < input_length) {
        int keyIndex = id % 4;
        char *keyCharPtr = (char *)&key;
        char keyChar = keyCharPtr[keyIndex];
        possible_plaintext_str_cuda[id] =
            keyChar ^ input_str_cuda[id];
    }
}
```

(e) Gemini Ultra.

```
__global__ void kernelXor(unsigned int key,
    char* input_str_cuda, unsigned char*
    possible_plaintext_str_cuda, int
    input_length) {
    int id = threadIdx.x + blockIdx.x *
        blockDim.x;
    if (id < input_length) {
        int keyIndex = id % 4;
        char* keyCharPtr = ((char*)&key);
        char keyChar = keyCharPtr[keyIndex];
        possible_plaintext_str_cuda[id] =
            keyChar ^ input_str_cuda[id];
    }
}
```

(f) GPT4.

Figure 10: Comparison of the generated kernelXor CUDA kernel.

```
void pow_cpu(int N, float ALPHA, float* X, int
    INCX, float* Y, int INCY) {
    int i;

    for(i = 0; i < N; ++i) {
        Y[i * INCY] = pow(X[i * INCX], ALPHA);
    }
}
```

(a) C++ code.

```
__global__ void pow_kernel(int N, float ALPHA,
    float *X, int INCX, float *Y, int INCY) {
    int i = (blockIdx.x + blockIdx.y *
        gridDim.x) * blockDim.x + threadIdx.x;

    if (i < N) {
        Y[i * INCY] = powf(X[i * INCX], ALPHA);
    }
}
```

(b) CUDA reference.

```
__global__ void pow_gpu(int N, float ALPHA,
    float *X, int INCX, float *Y, int INCY) {
    int i = (blockIdx.x + blockIdx.y *
        gridDim.x) * blockDim.x + threadIdx.x;
    if (i < N) {
        Y[i * INCY] = pow(X[i * INCX], ALPHA);
    }
}
```

(c) CODEROSETTA.

```
__global__ void pow_gpu(int N, float ALPHA,
    float *X, int INCX, float *Y, int INCY) {
    int i = blockIdx.x * blockDim.x +
        threadIdx.x;
    if (i < N) {
        Y[i * INCY] = powf(X[i * INCX], ALPHA);
    }
}

void pow_cpu(int N, float ALPHA, float *X, int
    INCX, float *Y, int INCY) {
    int threadsPerBlock = 256;
    int blocksPerGrid = (N + threadsPerBlock -
        1) / threadsPerBlock;
    pow_gpu<<<blocksPerGrid,
        threadsPerBlock>>>(N, ALPHA, X, INCX, Y,
        INCY);
    cudaDeviceSynchronize();
}
```

(d) GPT4.

```
// CUDA program to calculate the power of each element of an array

__global__ void pow_gpu(int N, float ALPHA, float *X, int INCX, float *Y, int INCY) {
    int i = blockIdx.x * blockDim.x + threadIdx.x;
    if (i < N) {
        Y[i * INCY] = powf(X[i * INCX], ALPHA);
    }
}

int main() {
    int N; // Number of elements in the array
    float ALPHA; // Exponent to raise each element to
    float *X; // Input array
    int INCX; // Stride of the input array
    float *Y; // Output array
    int INCY; // Stride of the output array

    // Initialize the input and output arrays

    // Allocate memory for the input and output arrays on the device

    // Copy the input array to the device

    // Call the kernel to compute the power of each element of the array

    // Copy the output array back to the host

    // Free the device memory

    return 0;
}
```

(e) Gemini Ultra.

Figure 11: Power of elements CUDA kernel.

```fortran
subroutine add
    use sp_data
    implicit none
    integer :: i, j, k, m
    if (timeron) call timer_start(t_add)
    do k = 1, nz2
        do j = 1, ny2
            do i = 1, nx2
                do m = 1, 5
                    u(m, i, j, k) = u(m, i, j,
                            k) + rhs(m, i, j, k)
                end do
            end do
        end do
    end do
    if (timeron) call timer_stop(t_add)
    return
end subroutine add
```

(a) Fortran code.

```cpp
void add() {
    int i, j, k, m;
    int thread_id = omp_get_thread_num();
    if (timeron && thread_id == 0) {
        timer_start(T_ADD);
    }
    #pragma omp for
    for (k = 1; k <= nz2; k++) {
        for (j = 1; j <= ny2; j++) {
            for (i = 1; i <= nx2; i++) {
                for (m = 0; m < 5; m++) {
                    u[k][j][i][m] =
                            u[k][j][i][m] +
                            rhs[k][j][i][m];
                }
            }
        }
    }
    if (timeron && thread_id == 0) {
        timer_stop(T_ADD);
    }
}
```

(b) C++ reference.

```cpp
void add() {
    int i, j, k, m;
    int thread_id = omp_get_thread_num();
    if (timeron && thread_id == 0) {
        timer_start(T_ADD);
    }
    #pragma omp for
    for (k = 1; k <= nz2; k++) {
        for (j = 1; j <= ny2; j++) {
            for (i = 1; i <= nx2; i++) {
                for (m = 0; m < 5; m++) {
                    u[k][j][i][m] =
                            u[k][j][i][m] +
                            rhs[k][j][i][m];
                }
            }
        }
    }
    if (timeron && thread_id == 0) {
        timer_stop(T_ADD);
    }
}
```

(c) CODEROSETTA.

```cpp
#include "sp_data.h"
void add() {
    int i, j, k, m;

    if (timeron) {
        timer_start(t_add);
    }
    for (k = 1; k <= nz2; k++) {
        for (j = 1; j <= ny2; j++) {
            for (i = 1; i <= nx2; i++) {
                for (m = 1; m <= 5; m++) {
                    u[m][i][j][k] =
                            u[m][i][j][k] +
                            rhs[m][i][j][k];
                }
            }
        }
    }
    if (timeron) {
        timer_stop(t_add);
    }
}
```

(d) GPT4.

```cpp
void add() {
    using namespace sp_data;
    int i, j, k, m;
    if (timeron) {
        timer_start(t_add);
    }
    for (k = 1; k <= nz2; k++) {
        for (j = 1; j <= ny2; j++) {
            for (i = 1; i <= nx2; i++) {
                for (m = 1; m <= 5; m++) {
                    u(m, i, j, k) = u(m, i, j, k) + rhs(m, i, j, k);
                }
            }
        }
    }
    if (timeron) {
        timer_stop(t_add);
    }
    return;
}
```

(e) Gemini Ultra.

Figure 12: Element-wise addition of two multi-dimensional arrays.

```
__global__ void set_valid_mask_gpu(const T
    *score, T score_thr, int *valid_mask, int
    dims) {
    int tid = blockIdx.x * blockDim.x +
        threadIdx.x;
    if (tid < dims) {
        if (score[tid] > score_thr) {
            valid_mask[tid] = 1;
        } else {
            valid_mask[tid] = 0;
        }
    }
}
```

(a) Generated CUDA code.

```
template <typename T>
__global__ void set_valid_mask_gpu(const T
    *score, T score_thr, int *valid_mask, int
    dims) {
    int tid = blockIdx.x * blockDim.x +
        threadIdx.x;
    if (tid < dims) {
        if (score[tid] > score_thr) {
            valid_mask[tid] = 1;
        } else {
            valid_mask[tid] = 0;
        }
    }
}
```

(b) Corrected CUDA code.

```
__global__ void get_ev(double *old_arr, double
    *new_arr) {
    int tid = blockIdx.x * blockDim.x +
        threadIdx.x;
    if (tid < size) {
        new_arr[tid] = old_arr[tid];
    }
}
```

(c) Generated CUDA code

```
__global__ void get_ev(double *old_arr, double
    *new_arr, int size) {
    int tid = blockIdx.x * blockDim.x +
        threadIdx.x;
    if (tid < size) {
        new_arr[tid] = old_arr[tid];
    }
}
```

(d) Corrected CUDA code

Figure 13: Post Compilation fixes on CUDA kernel.

```
static void makexgraph(graph *g, xword *h, int
    n) {
    setword gi;
    int i, j;
    xword hi;
    for (i = 0; i < n; ++i) {
        hi = 0;
        gi = g[i];
        while (gi) {
            j = FIRSTBITNZ(gi);
            gi ^= bit[j];
            hi |= XBIT(j);
        }
        h[i] = hi;
    }
}
```

(a) Input C++ code.

```
__global__ void makexgraph(graph *g, xword *h,
    int n) {
    setword gi;
    int i = blockIdx.x * blockDim.x +
        threadIdx.x;
    xword hi;
    for (; i < n; i += blockDim.x * gridDim.x)
        {
        hi = 0;
        gi = g[i];
        while (gi) {
            j = FIRSTBITNZ(gi);
            gi ^= bit[j];
            hi |= XBIT(j);
        }
        h[i] = hi;
    }
}
```

(b) Intermediate CUDA generated code.

Figure 14: Back translation intermediate results.

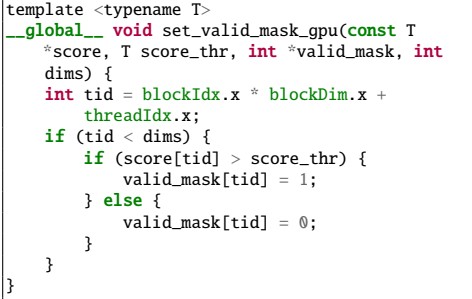

```
You are an expert in translating C++ programs to CUDA programs.
Given the C++ program below, translate it to CUDA. Ensure that the CUDA
↪   program is compatible with the C++ program and preserves the
↪   semantics of the original code.
Just print the CUDA program and remove any unnecessary comments. Surround
↪   the generated CUDA program in #start and #end.

### C++ Program:{cpp_code_content}

### CUDA Version:
```

Figure 15: Prompt for translating C++ to CUDA.

```
void initializeArray(std::unique_ptr<int[]>&
    array, int size) {
    for (int i = 0; i < size; ++i) {
        array[i] = i * 10;
    }
}
```

(a) Example of C++ code with `std::unique_ptr`

```
__global__ void initializeArray(int *array, int
    size) {
    int i = blockIdx.x * blockDim.x +
        threadIdx.x;
    if (i < size) {
        array[i] = i * 10;
    }
}
```

(b) Translated CUDA code.

Figure 16: Example of translation of a C++ code with `std::unique_ptr`

```
void getRho(const int numOfNucl, const double*
    psi, const double* occNo, double* rho,
    const char debug) {
    *rho = 0;
    for (int i = 0; i < numOfNucl; ++i) {
        *rho += occNo[i] * psi[i] * psi[i];
    }
    if (debug == 1) {
        printf("DEBUG _ print _ of _ RHO:\n_
            RHO _ = _ %f\nThis _ is _ the _
            last _ line (RHO).\n\n", *rho);
    }
}
```

(a) C++ Code

```
__global__ void getRho(const int numOfNucl,
    const double *psi, const double *occNo,
    double *rho, const char debug) {
    *rho = 0;
    for (int i = blockIdx.x * blockDim.x +
        threadIdx.x; i < numOfNucl; i +=
        blockDim.x * gridDim.x) {
        *rho += occNo[i] * psi[i] * psi[i];
    }
    if (debug == 1) {
        printf("DEBUG print of RHO:\nRHO =
            %f\nThis is the last line
            (RHO).\n\n", *rho);
    }
}
```

(b) Wrong translated CUDA code.

Figure 17: Example of a failed C++ to CUDA translation.

