# OpenReview forum: "CodeRosetta: Pushing the Boundaries of Unsupervised Code Translation for Parallel Programming"
_NeurIPS.cc/2024/Conference — NeurIPS 2024 poster_

### Official Review · Reviewer_BqwA · 2024-07-07

**Soundness:** 2
**Presentation:** 2
**Contribution:** 2
**Rating:** 6
**Confidence:** 4

**Summary:**

This work intorduces an encoder-decoder transformer model, CodeRosetta, translating between programming languages and also their high-performance computing (HPC) extensions parallelly.

The authors claim that CodeRosetta outperforms baselines and general LLMs on C++ to CUDA and Fortran translation tasks.

**Strengths:**

1. In the translation task from C++ to CUDA and Fortran, the paper claims that the CodeRosetta 0.8B model outperforms much larger language models, including GPT-4, Gemini-Ultra, and StarCoder 15B.
2. This research employed various techniques, such as Abstract Syntax Tree (AST) Entity Recognition, Noise Injection, Token Dropping, and Language-specific Token Insertion, combined with supervised fine-tuning to enhance the model's performance on the code translation task.

**Weaknesses:**

1. Both BLEU and CodeBLEU might not serve as accurate translation performance indicators [1, 2]. These semantic metrics struggle to capture the nuanced behaviors of code. While Compilation Accuracy appears to be a robust metric, it lacks a formal definition in the paper.
2. The functional consistency of the original code and the translated code is more important than whether the translated code is compilable. Adding functional correctness evaluation may make the experiment sound more reliable.
3. For the C++ to CUDA translation task, the paper notes that the model is trained on an unpaired collection of C++ and CUDA source files. This training method can explain how the model learns the knowledge of each individual language. However, a more detailed explanation is needed to clarify the source of the model's language translation capability.

[1] Evtikhiev, M., Bogomolov, E., Sokolov, Y., & Bryksin, T. (2023). Out of the bleu: how should we assess quality of the code generation models?. *Journal of Systems and Software*, *203*, 111741.

[2] Chen, M., Tworek, J., Jun, H., Yuan, Q., Pinto, H. P. D. O., Kaplan, J., ... & Zaremba, W. (2021). Evaluating large language models trained on code. *arXiv preprint arXiv:2107.03374*.

**Questions:**

I am unfamiliar with the programming language compilers, but AST conversion should vary for C, CUDA, and Fortran. CodeRosetta may be unable to distinguish different languages and much less to generate executable code? Additionally, finetuning with synthetic data from language models can be considered as supervised learning, which may conflict with the paper topic?

**Limitations:**

The authors addressed limitations in their work.

---

> ### Author Rebuttal · Authors · 2024-08-07
>
> **Metric issue and Formal definition of compilation**
>
> We understand the metrics used may not be comprehensive enough, as noted in [1]. To address this, we manually analyzed and executed the translated code. Please reference to global response.
> Moreover, as it was mentioned in Out of the BLEU [1] that ChrF is a better fit for evaluating the generated code, we calculated the results based on this metric. The results on ChrF show that CodeRosetta achieves a good score in comparison to other models. Note that we omitted BabelTower from the baseline as neither its model nor its results are available.
>
> | Model      | CodeRosetta | GPT4  | Gemini Ultra | Gemini Pro |
> |------------|-------------|-------|--------------|------------|
> | ChrF(CUDA) | 81.05       | 70.15 | 73.20        | 72.58      |
>
> Compilation Accuracy is the ratio of compilable generated code to the total number of reference codes in the test set.
>
> **Functional correctness**
>
> Please refer to the global response.
>
> **Details on the model's language translation capability.**
>
> The bilingualism of our model stems from several pre-training and training objectives that we have utilized and developed. For example, the Mask Language Modeling helps the model to learn cross-lingual representation. For instance, when a `if` keyword, which is a shared keyword between C++ and Fortran, is masked, forces the model to learn the context of an if statement; that is how such a statement looks like in each language. On the other hand, AST entity recognition teaches the model about language-specific entities and cross-language entities to further enable the model to map similar programs in different languages to the same embedding space (encoder model). Then, Denoising Auto Encoding and Back Translation enable the model to leverage the embeddings learned in previous steps to train an encoder-decoder model in order to generate code in the target language.
>
>
> **Fine Tuning with synthetic data**
>
> Finetuning is an optional step. One of the biggest challenges in training a code translation model is the lack of paired data. Therefore, a common practice is to train a model in an unsupervised way. However, even if paired data is available, the amount of data could be insufficient to train a model from scratch. Here, we aimed to evaluate how a model will perform if a small set of paired data exists for fine-tuning. Since the model already has knowledge of code translation, this fine-tuning step can further improve the result of the model and make its performance comparable to a much larger model. CodeRosetta is one of the few unsupervised trained models that support fine-tuning if the data exists.
>
> **Differences in ASTs of different languages**
>
> That is indeed true. The ASTs differ slightly depending on which language we are dealing with. However, some entities are common between ASTs, such as identifiers. These common entities enable the model to learn how they are utilized in the source code. For example, how identifiers are being used in C++ and CUDA programs. Some other entities are not common among the languages, such as pointers. The absence of such an entity in a programming language forces the model to not predict the type of an entity as a pointer. This, on the other hand, helps the model to realize that pointers are available in a programming language but are not available in another language.

---

> > ### Comment · Reviewer_BqwA · 2024-08-10
> > **Response to Author Rebuttal**
> >
> > Dear Authors,
> >
> > Thank you for your prompt response.
> >
> > **Metric Issue and Formal Definition of Compilation**
> >
> > * **ChrF Metric:** I assume 'ChrF' refers to the 'Character n-gram F-score'. Could you share the $\beta$ value used in your settings? Much like token-level metrics, the ChrF metric at the character level may also fail to accurately reflect the true code translation capabilities.
> >
> > * **Compilation Accuracy:** In natural language translation tasks, language models often generate specious responses. I am concerned that your model might generate code that compiles correctly but is functionally different from the intended solution.
> >
> > **Functional Correctness**
> >
> > * I understand that it is not feasible to conduct extensive experiments to evaluate the functional correctness of the generated code within the rebuttal period. I appreciate your efforts in manually analyzing the model's outcomes.
> >
> > **Fine-Tuning with Synthetic Data**
> >
> > * Could you clarify whether the scores reported in Tables 1 and 2 originate from the pre-trained model or the fine-tuned model?
> >
> > **Model’s Language Translation Capability**
> >
> > * Given that you are pre-training the model with unsupervised objectives, I recognize that the model can acquire monolingual capabilities, and perhaps some bilingual capabilities from frequently occurring common usages—the area where rule-based deterministic algorithms excel. However, I am curious about the model's ability to handle novel cases. For example, you mentioned that "some entities are not common among languages, such as pointers." How does the model handle the conversion of 'cuda::unique_ptr' to C++ code?
> >
> > **Differences in ASTs of Different Languages**
> >
> > * Thank you for providing an answer to my question.

---

> ### Author Response · Authors · 2024-08-11
> **Thank you! --- Part 1**
>
> Dear Reviewer BqwA,
>
> Thank you for your quick response, your thoughtful feedback, and the opportunity to address your questions.
>
> ---
>
> ## Metrics and Compilation Definition
>
> **ChrF Metric:** Yes, ChrF refers to the Character n-gram F-score. We apologize for not providing the full name earlier. We used the HuggingFace Evaluate Metric library with the default $\beta$ value set to 2. We used the ChrF metric following the recommendations from the *“Out of the BLEU: How Should We Assess Quality of the Code Generation Models?”* paper, which states that: *``ChrF and ROUGE-L are the best-performing metrics for the assessment of code generation models among the metrics we consider.``* We have also included the results  using the ROUGE-L metric:
>
> | Model         | CodeRosetta | GPT4  | Gemini Ultra | Gemini Pro |
> |---------------|-------------|-------|--------------|------------|
> | Rouge-L(CUDA) | 82.12       | 63.37 | 69.27        | 69.82      |
>
>
> **Compilation Accuracy:** We acknowledge the limitation of using compilation accuracy as a metric in code translation, as it is possible for a model to generate code that compiles successfully but diverges from the intended functionality. To partially address this, we conducted a manual evaluation of 30 generated code samples to compare their behavior against reference implementations. While we fully understand the limitations of such evaluations, we found that the functional correctness of generated code was preserved in the majority of samples (~93%). We also did run those generated codes against the reference and found them to be compilable and functionally correct producing the same outputs with the same inputs as the reference. In addition, the use of the compilation accuracy metric enabled us a direct comparison with BabelTower (ICML 2022), especially given that their codebase is not open sourced.
>
> ---
>
> Following this discussion, we will dedicate a section in our paper that covers the limitations of existing metrics in code translation and our efforts to partially address these limitations through manual inspection. We will also include the samples in the Appendix.
>
> Taking all of these additional results into account, we believe we have made our best effort to address this concern, though we acknowledge it may not be perfect. That said, we would be more than happy to include any additional metrics you suggest that may better characterize the performance of our work.
>
> ---
>
> ## Fine-Tuning
>
> `Paper-Tables-1 & 2` report the results from the fine-tuned model. `Paper-Table-3-Page-9` reports the results from the pre-trained model.
>
> **Discussion on Fine-tuning for Code Translation w/o Verification in Supervised Manner.** In code translation, paired data is scarce. However, our model benefits from a foundational understanding of code translation acquired through unsupervised and self-supervised pre-training (243K training examples). We show that finetuning, even on limited synthetic data—*without verifying the generated samples and their one-to-one mapping to input code in a supervised manner*—generated by larger models (merely 5K paired samples---less than 2% of total data points), can further boost the model’s performance. While synthetic data may introduce some errors (as large models can make mistakes in their translation), the combination of foundational understanding during pre-training and fine-tuning with such a small number of synthetic data points can lead to additional improvements.

---

> ### Author Response · Authors · 2024-08-11
> **Thank you! --- Part 2**
>
> ## Model’s Translation Capability
>
> Thanks for this great question. As you mentioned, there are entities, libraries, keywords, and syntaxes from the source language (e.g., C++) that may not be valid in the target language (e.g., CUDA). `std::unique_ptr` and other C++ Standard Template Libraries (STL) belong to this category and must be avoided in the translation. The pre-training process in CodeRosetta (esp. Weighted Token Dropping) equips the model to gain a semantic understanding of source and target languages. It reduces the occurrences of invalid tokens in the translation. Nonetheless, there may be cases in which the model fails to correctly translate from common entities in the source code to valid entities in the target code.
>
> Looking into our dataset, there was no occurrence of `std::unique_ptr` in the test set. However, as the training set has `std::unique_ptr`, we intended to test the capability of CodeRosetta by developing a C++ code that uses `std::unique_ptr`, we made sure this sample does not exist in the training set:
>
> ### Reference C++ Code
>
> ```
> void initializeArray(std::unique_ptr<int[]>& array, int size) {
>     for (int i = 0; i < size; ++i) {
>         array[i] = i * 10;
>     }
> }
>
> ```
>
> ### CodeRosetta CUDA Generation
>
> ```
> __global__ void initializeArray(int *array, int size) {
>     int i = blockIdx.x * blockDim.x + threadIdx.x;
>     if (i < size) {
>         array[i] = i * 10;
>     }
> }
> ```
>
> This sample is about array initialization. As can be seen, the model is able to provide the CUDA code. It removes the `std::unique_ptr` from the CUDA kernel as it is not supported in the device code. In CUDA, `std::unique_ptr` is called in the host kernel, which is the main function. Our model focuses on the generation of device function and hence is successful in this case.
>
> Moreover, from the test set, we checked where our model could potentially fail. Here is an example of a translation that CodeRosetta struggles with:
>
> ### Reference C++ Code:
> ```
> void getRho(const int numOfNucl, const double* psi, const double* occNo, double* rho, const char debug) {
>     *rho = 0;
>     for (int i = 0; i < numOfNucl; ++i) {
>         *rho += occNo[i] * psi[i] * psi[i];
>     }
>     if (debug == 1) {
>         printf("DEBUG ▁ print ▁ of ▁ RHO:\n▁ RHO ▁ = ▁ %f\nThis ▁ is ▁ the ▁ last ▁ line (RHO).\n\n", *rho);
>     }
> }
> ```
> ### CodeRosetta CUDA Generation:
>
> ```
> __global__ void getRho(const int numOfNucl, const double *psi, const double *occNo, double *rho, const char debug) {
>     *rho = 0;
>     for (int i = blockIdx.x * blockDim.x + threadIdx.x; i < numOfNucl; i += blockDim.x * gridDim.x) {
>         *rho += occNo[i] * psi[i] * psi[i];
>     }
>     if (debug == 1) {
>         printf("DEBUG print of RHO:\nRHO = %f\nThis is the last line (RHO).\n\n", *rho);
>     }
> }
> ```
>
> In this example, CodeRosetta, along with all larger models like GPT-4, Gemini Ultra, and Gemini Pro, failed to translate the code correctly. We verified the generated code by executing it. The line `*rho = 0;` initializes the variable rho to zero. However, if this kernel is executed by multiple threads across different blocks simultaneously (which is very likely), all threads will write to the same memory location. This results in a race condition where multiple threads are simultaneously updating the value of *rho without any synchronization mechanism, such as atomic operations or reduction techniques. As a result, each thread's write could overwrite others, leading to incorrect and unpredictable outcomes. To rectify this issue, `rho` should be set to 0 in the host code, and `atomicAdd` should be included in the computation.
>
> ---
>
> We hope these clarifications address your feedback and questions effectively and you find our rebuttal and additional discussion satisfactory.
>
> We would greatly appreciate it if these additional results and clarifications lead you to reevaluate our work.
>
> At the end, if there are any other lingering questions or requests, please do not hesitate to ask. Our goal is to work with experts like you to improve the quality of our paper and make our research more valuable for the community.
>
> Thank you,
>
> The Authors

---

> > ### Comment · Reviewer_BqwA · 2024-08-11
> > **Thank you for your effort**
> >
> > Dear authors,
> >
> > Thank you for your diligent effort and thorough response!
> >
> > The experiments and explanations you provided have satisfactorily addressed all of my concerns. As a result, I am pleased to elevate my score from 3 to 6 and advocate for the paper's acceptance.
> >
> > Good luck!

---

> > > ### Author Response · Authors · 2024-08-11
> > > **Thank you for your time, thoughtful engagement, and recognition of our efforts!**
> > >
> > > Dear Reviewer BqwA,
> > >
> > > Thank you for your thoughtful engagement and for taking time to carefully review our response. It means a lot to us. We appreciate your recognition of our efforts and are delighted that our explanations and additional experiments have addressed your concerns.
> > >
> > > We're especially grateful for your decision to increase the score and advocate for the acceptance of our paper. Your feedback has improved the quality of our work, and we believe this will benefit the broader research community.
> > >
> > > We are committed to revising the manuscript for the final version to carefully reflect our discussions.
> > >
> > > Thank you once again for your support.
> > >
> > > Best Regards,
> > >
> > > The Authors

---

### Official Review · Reviewer_W1wi · 2024-07-08

**Soundness:** 3
**Presentation:** 3
**Contribution:** 3
**Rating:** 5
**Confidence:** 4

**Summary:**

This paper introduces CodeRosetta, an encoder-decoder transformer model for unsupervised translation between programming languages and their high-performance computing (HPC) extensions. The main contributions include unsupervised code translation: CodeRosetta translates between programming languages (e.g., C++) and their parallel programming extensions (e.g., CUDA and Fortran). Customized pre-training and training objective: The model employs Abstract Syntax Tree (AST) Entity Recognition and customized Denoising Auto-Encoding (DAE) to learn parallel programming syntax and nuances.

Experimental results demonstrate that CodeRosetta outperforms state-of-the-art methods in C++ to CUDA translation, with improvements of 2.9 BLEU and 1.72 CodeBLEU points, and a 6.05% increase in compilation accuracy.

**Strengths:**

1. This paper introduces two new learning objectives: Abstract Syntax Tree (AST) Entity Recognition (AER) and customized Denoising Auto-Encoding (DAE) with weighted token dropping and insertion.

2. CodeRosetta can learn both the general syntactic structure of code and the specific nuances of parallel programming constructs without relying on language-specific metrics.

3. CodeRosetta outperforms the current state-of-the-art baseline models.

**Weaknesses:**

- The evaluation only included BLEU and CodeBLEU, which are syntax-based metrics, and compilation correctness. It did not test the runtime correctness of the code, indicating a lack of completeness in the evaluation metrics.

- In Section 5.2 Ablation Study, the ablation study did not include parts about Masked Language Modeling (MLM) and back translation.

**Questions:**

- In Section 5.2 Ablation Study, Table 3: Ablation Study for C++ to CUDA, it was found that removing the fine-tuning with data from large models resulted in the most significant performance drop. Although the authors stated that this step is optional, the model's performance without this step did not surpass the baseline.

- Section 3.4 Back translation for unsupervised refinement and Section 3.5 Finetuning with synthetic data from language models both mention generating new data using large models. There is a potential risk of data leakage during this process. It is crucial to ensure that the data generated by large models does not overlap with the test data, which could affect the validity of the test results.

- Minor issues.  Line 27,  "C++ ↔ CUDA or C++ ↔ CUDA"  should be "C++ ↔ CUDA or C++ ↔ Fortran"?

**Limitations:**

The authors have addressed the limitations of their approach.

---

> ### Author Rebuttal · Authors · 2024-08-07
>
> **Evaluation metrics and runtime correctness**
>
> Evaluation metrics were selected for comparison as they have been utilized in the baselines. We understand that these metrics can have limitations. We tried to address this point by manually analyzing the translated code and executing it. Please refer to the global response for further details on this. Thank you.
>
> **MLM and BT in ablation**
>
> We have performed an ablation study on the parts that are contributions of our work. The intention was to show the contribution of each proposed training objective. Technically, there is no issue with performing an ablation study on MLM and BT. We make sure to add them to the paper per your request. However, since this requires retraining the model two more times, it would not be possible to finish it within the rebuttal time constraint.
>
> **Fine-tuning Performance**
>
> We have added fine-tuning as an additional optional step to our pipeline.
> For translation works, one of the biggest obstacles is the lack of paired data corpora. For example, it is very challenging to find a C++ program and its equivalent CUDA code. However, even if such data exists, it would not be sufficient enough to train a model. Therefore, developing an unsupervised translation model is inevitable. One of the benefits of having such a model is that we can train it using self-supervised and unsupervised training techniques as proposed in the paper and then fine-tune it on a small set of paired data. This is a key ingredient as the model already has knowledge of translation, and with a small set of paired data, it can gain a significant boost in its performance.
>
> **Contamination/Deduplication for synthetic data:**
>
> During Back Translation (BT), no new model or training data is involved. In this training task, only CodeRosetta model and the train set are involved. CodeRosetta is asked to translate from A to B. However, since the model has not translated any code in the past, B would be of poor quality with noises. Then B is used as input to the model to reconstruct A. So, the model leverages weak supervision from its own knowledge, and the test set is not involved at all.
> For synthetic, we take random samples only from the train set and ask a larger model to translate them.
> To investigate this further, we applied CodeBERTScore to measure the functional similarity of each sample in the test set against the synthetic dataset.
> And randomly analyzed some samples that have high CodeBERTScore.
>
> For example, this is one of the samples from test set
> ```
> __global__ void l1_kernel(int n, float *pred, float *truth, float *delta) {
>     int i = blockIdx.x * blockDim.x + threadIdx.x;
>     if (i < n) {
>         float diff = pred[i] - truth[i];
>         delta[i] = (diff > 0) ? 1 : -1;
>     }
> }
> ```
> The most similar CUDA code in the trainset with CodeBERTScore of 0.92 is this:
>
> ```
> __global__ void softmax_x_ent_cpu(int n, float *pred, float *truth, float *delta, float *error) {
>     int i = blockIdx.x * blockDim.x + threadIdx.x;
>     if (i < n) {
>         float t = truth[i];
>         float p = pred[i];
>         error[i] = (t) ? -log(p) : 0;
>         delta[i] = t - p;
>     }
> }
> ```
>
> As we can see, the only common thing about these kernels is the fact that both of them are loss functions. However, the first kernel is L1 loss, and the second kernel is cross-entropy loss.
>
> Here, we are showing the amount of data that falls in each range of similarity score. The ranges with zero data are omitted.
>
> | 0.4-0.5 | 0.5-0.6 | 0.6-0.7 | 0.7-0.8 | 0.8-0.9 | 0.9-1.0 |
> |---------|---------|---------|---------|---------|---------|
> | 0%      | 0.8%    | 33%     | 58%     | 7%      | 0.05%   |
>
> A score below 0.8 shows moderate to small similarity. As can be seen, the majority of the files don’t have high similarity with the test data, and even with those ones that had high similarity, we checked manually and realized the reason for the high similarity is that they belong to the same domain with slight similarity.

---

### Official Review · Reviewer_djHa · 2024-07-09

**Soundness:** 2
**Presentation:** 2
**Contribution:** 2
**Rating:** 5
**Confidence:** 4

**Summary:**

The paper presents CodeRosetta, a transformer model designed for unsupervised code translation and their high-performance computing extensions, such as C++ to CUDA and Fortran. By introducing novel pre-training objectives like AST Entity Recognition and customized Denoising Auto-Encoding, CodeRosetta effectively learns parallel programming syntax. The model demonstrates superior performance, surpassing state-of-the-art baselines metrics like BLEU / CodeBLEU and compilation accuracy.

**Strengths:**

**Strengths**

1. CodeRosetta introduces novel pre-training objectives to solve translation tasks about CUDA and Fortran, such as AST Entity Recognition and customized Denoising Auto-Encoding.
2. CodeRosetta is one of the first to handle code translation of code HPC extensions, marking an advancement in code intelligence.
3. The model outperforms state-of-the-art baselines in key metrics, achieving higher BLEU and CodeBLEU scores

**Weaknesses:**

**Weaknesses**

1. Although novel pre-training objectives have been introduced, the authors have not methodologically distinguished these from predecessors using AST or Denoising objects. Training with code structures is prevalent in code learning, as seen in models like CodeT5[1] and PLBART[2]. The authors should discuss these previous methods and variants of structural modeling, as referenced in papers like [3,4].
2. The experimental section lacks some rigor. For instance, in the Educational value filtering, "randomly sampled 100,000 C++ files from Stack V2 and employed GPT-3.5 to assess their ‘educational value’ for learning C++ coding concepts" poses challenges to reproducibility. The use of GPT-3.5 to evaluate C++ coding concepts and the prompt engineering methodology warrant further clarification. Additionally, some experimental details in Table 4 are not clearly presented.
3. Upon reviewing the data composition of models like StarCoder (StarCoderDataV1), which includes 58,355 CUDA and 165,446 Fortran files, it appears that existing models like StarCoder or DeepSeekCoder could be directly employed for research. Significant engineering effort in the paper was focused on vocabulary issues, such as inserting CUDA-specific tokens, yet there is no discussion on how existing code LLMs handle out-of-vocabulary problems with languages like CUDA.

[1] CodeT5: Identifier-aware Unified Pre-trained Encoder-Decoder Models for Code Understanding and Generation

[2] Unified Pre-training for Program Understanding and Generation

[3] A Survey of Neural Code Intelligence: Paradigms, Advances and Beyond

[4] A Survey on Pretrained Language Models for Neural Code Intelligence

**Questions:**

1. Is there an error in the left part of Figure 4?
2. The phrase "compare CodeRosetta against their best fine-tuned model, StarCoder (15B)" is unclear. While StarCoder's pre-training data does include Fortran code, the notion of "best fine-tuned" is not well explained. Could the authors clarify this?
3. The performance of the three closed-source models in Table 4 is surprisingly low. Could the authors provide more details on the testing procedures?

**Limitations:**

I do not identify potential negative societal impacts in this paper.

---

> ### Author Rebuttal · Authors · 2024-08-07
>
> **Difference with relevant works, CodeT5[1] and PLBART[2]**
>
> Thank you for providing us with these references. Indeed, Denoising Auto Encoding (DAE) is a popular technique when it comes to training Encoder-Decoder models, as both CodeT5 and PLBART use it, too. Some of the noising strategies, such as masking and token dropping, are common. One of the key differences in the noising strategies employed by CodeRosetta is their language-specific characteristics. For example, instead of random token dropping, we employ weighted random dropping that gives a higher probability to language-specific reserved keywords, forcing the model to develop a deeper understanding of the target language’s semantics and its structure. Another noising strategy is token insertion, which encourages the model to distinguish between valid and invalid tokens for the target language. We make sure to update the paper and distinguish CodeRosetta’s noising strategies vs. common strategies in the literature, specifically in the related works section.
>
> **Details on filtering CPP files from StackV2**
>
> When training CodeRosetta, We aim to present the same amount of data for the languages involved in the translation. For the translation of Fortran to C++, we used data from the StackV2 data corpus, which contains far more C++  than Fortran. It is possible to randomly subsample C++ source code files to have an equal number of samples for C++ and Fortran. However, instead of random subsampling, we tried to extract C++ samples that have ‘good quality’. We employ a technique similar to “Textbooks are all you need”[1]. Using a larger model to assess the quality of some portion of C++ data in StackV2 (due to the budget) and then training a classifier model to classify all C++ data in StackV2, then randomly selecting from the C++ files that have been predicted to have good educational value. We make sure to rephrase this part of the paper and provide more clear explanations with references. Thank you.
>
> [1] S. Gunasekar, Y. Zhang, J. Aneja, C. C. T. Mendes, A. Del Giorno, S. Gopi, M. Javaheripi, P. Kauffmann, G. de Rosa, O. Saarikivi et al.,“Textbooks are all you need,” arXiv preprint arXiv:2306.11644, 2023`
>
>
>
> **Clarification on Table 4**
>
> The table presents Fortran to C++ translation results (apologies for the mistake in the caption of the title; we make sure to rectify its title) on the dataset provided by Bin et al. [9]. In this dataset, the test set contains 33 paired samples, meaning for each Fortran code there exists an equivalent C++ code. We aim to analyze the performance of CodeRosetta for the task of translating from Fortran code to C++ code. The experimental results in this table indicate that for the Fortran to C++ translation, CodeRosetta is effective.
>
> **Comparison with StarCoder or DeepSeekCoder**
>
> Thank you for suggesting a comparison with StarCoder and DeepSeekCoder.
> We have provided the results in the following table using the prompt format provided in Figure 6 of the paper, with adjustments depending on which languages we are targeting. Results indicate that StarCoder is relatively better than DeepSeekCoder, and it has a CodeBLEU score close to Gemini-Ultra. We make sure to update the paper and reflect the result of these open Code LLMs.
>
> |                              |       | C++ to CUDA |      | Fortran to C++ |
> |------------------------------|-------|-------------|------|----------------|
> |                              | BLEU  | CodeBLEU    | BLEU | CodeBLEU       |
> | DeepSeek-Coder-V2-Lite-Base (16B)  | 26.63 | 21.46       | 0.77 | 12.09          |
> | Starcoder2-15b-instruct-v0.1 | 37.58 | 62.58       | 5.71 | 18.21          |
>
>
> **Fine-tuned StarCoder(15B)**
>
> StarCoder has been fine-tuned for the task of C++ to Fortran Translation by Bin *et al.* [9]. They have fine-tuned other models as well, among them, fine-tuned StarCoder was able to translate C++ to Fortran with higher accuracy. We compared CodeRosetta against the fine-tuned StarCoder since this model was the one that had the best results among others.
>
>
> **Low scores of ChatGPT and Gemini**
>
> Baseline papers have used BLEU and CodeBLEU to evaluate the translation results. We have used the same metrics. However, LLMs generally do not perform one to one translation. They typically provide descriptions and instructions about code and how to execute it. This could impact the metrics. Moreover, to reveal why other LLMs' results are lower than CodeRosetta, we manually examined the generated code. Please refer to the global response for further explanation on this.

---

> ### Comment · Reviewer_djHa · 2024-08-13
> **Response from reviewer**
>
> Thank you for your rebuttal. My question was about Figure 4, not Table 4. My concerns have been largely addressed, and I have raised my evaluation.

---

> > ### Author Response · Authors · 2024-08-13
> > **Thank you for your time and recognition of our efforts!**
> >
> > Dear Reviewer djHa,
> >
> > Thank you for your thoughtful engagement with our work and for taking time to carefully read our rebuttal. We are glad to hear that our explanations and additional experiments have largely addressed your concerns, and we sincerely appreciate your decision to raise your evaluation.
> >
> > We apologize for the misunderstanding your point regarding Figure 4. Upon review, we identified a typo in the figure, and we have since corrected it to align with the text.
> >
> > We are committed to incorporating these additional results and discussions in the final version of the paper, ensuring that they accurately reflect our discussions.
> >
> > Thank you once again for your support and valuable feedback.
> >
> >
> > Best regards,
> >
> > The Authors

---

### Official Review · Reviewer_9e24 · 2024-07-13

**Soundness:** 3
**Presentation:** 3
**Contribution:** 4
**Rating:** 7
**Confidence:** 3

**Summary:**

The paper looks into the problem of unsupervised code translation with a particular focus on parallel programming languages like C++ to CUDA. It trains (relatively) small encoder-decoder models compared to current LLMs while achieving very strong performance on translation tasks. It incorporates program-specific terms in loss such as token frequencies, and AST structure to achieve this.

**Strengths:**

1. Significance - This is an impactful problem and the resulting approach competes with very large LLMs with ~small models. The authors perform appropriate ablations and the proposed AST structure and token frequency-based denoising in loss are also effective.

2. Clarity - The paper is well-written and provides appropriate context and background around methods.

3. Originality - While people have tried including program-specific constructs in loss previously for encoder models applying it to the translation domain in novel. The denoising scheme based on language-specific token frequencies is intuitive and is proven effective from thorough ablations

4. Qualitative analysis of compilation errors - It was good to see authors performing an analysis of mistakes in model-translated samples. The distribution or errors and thus motivated future direction of using execution feedback does look promising.

**Weaknesses:**

1. Evaluation and metrics. Perhaps a challenging aspect of the results is the evaluation benchmarks and datasets used.
  a. Benchmarks. The authors achieve considerably strong results on the C++ to CUDA evaluation sets with BLEU and CodeBLEU numbers higher than 75. Do authors perform some analysis on contamination or deduplication on the benchmark and collected training sets?
  b. Evaluation metric. Functional correctness has emerged to be the winning metric in the code generation space. However, it seems more challenging to apply that to the translation problems. High BLEU might not necessitate better translations. Similarly, comparability as a metric does not ensure that the translation is correct. While it might be hard to design functional correctness-based evaluation, the authors do not perform any other evaluation to assess the quality of translation beyond the, somewhat unreliable automated metrics.

2. Performance on proprietary LLMs. GPT-4 and Gemini-Pro depict considerably worse BLEU for some language pairs. It would be useful to highlight the failure modes since it might also highlight some issues in BLEU-like metrics.

3. Missing details on the datasets. The authors can provide more details on the datasets and benchmarks used in terms of document length and document counts. Similar statistics for the benchmarks and model generations would be useful to understand the complexity of the task.

**Questions:**

1. The AST structure loss seems interesting. Do the authors believe it would also improve embeddings for monolingual encoder models?

2. Backtranslation-based unsupervised learning. The authors use back-translation to improve the translation performance of the model. However, for given languages pairs A and B (going from A -> B -> A), if the intermediate sample B is not constrained, the model can cheat and not do the translation job currently (as an extreme example, just copy the tokens from A). Can the authors clarify how the setup avoids such failure models?

3. What tokenizers do the authors use for these languages which are usually less well represented in the web data?

**Limitations:**

Authors have limitations in the the appendix.

---

> ### Author Rebuttal · Authors · 2024-08-07
>
> **Deduplication analysis**
>
> The C++ to CUDA dataset was obtained from BabelTower [26]. This dataset has already gone through a rigorous deduplication and cleaning process. Moreover, there is no paired trained data available in the training set. This means the model can not see a C++ code and its CUDA equivalent during training as such data does not exist. The model has to rely on the self-supervised training objectives to learn to embed source code in different languages into the same embedding space. Only for the test set, we have paired data, which we have used to evaluate our model.
> We further evaluated the similarity between the test and the trainset from BabelTower [26] using CodeBERTScore. The following table shows the ranges for CodeBERT score and the amount of data that falls in each range. For example, 48.61% of training data have CodeBERTScore between 0.7 and 0.8 when compared against test data. Ranges with zero data are omitted. A score below 0.8 shows low or moderate similarity. As can be seen, the majority of samples in the training set have CodeBERTScore lower than 0.8
> | 0.4-0.5 | 0.5-0.6 | 0.6-0.7 | 0.7-0.8 | 0.8-0.9 | 0.9-1.0 |
> |---------|---------|---------|---------|---------|---------|
> | 0%      | 1.7%    | 44.80%  | 48.61%  | 4.78%   | 0.03    |
>
> We randomly examined the training samples that had high similarity scores.
> For example, this is a sample from the test set:
>
> ```__global__ void scale_dev(float *array, float scale, int N) {
>     int idx = blockIdx.x * blockDim.x + threadIdx.x;
>     if (idx < N) {
>         array[idx] *= scale;
>     }
>     return;
> }
> ```
>
> And the most similar sample from the trainset is this one with CodeBERTScore of 0.94:
>
> ```
> __global__ void set_val(float *data, float val, int size) {
>     int idx = blockIdx.x * blockDim.x + threadIdx.x;
>     if (idx >= size) return;
>     data[idx] = val;
> }
> ```
>
> We can see that despite some similarities between the two code snippets, their functionalities are different. One is scaling the values of an array; the other is populating an array.
>
> **Functional Correctness**
>
> Please refer to the global response.
>
> **More details on datasets**
>
> Details of the datasets have been added to the appendix. Please let us know if any other information needs to be added. We will make sure to move the datasets' table to the main paper if space permits.
>
>
> **AST for monolingual models**
>
> AST entity recognition (AER) pre-training task enables the model to identify various entities in the source code. Examples of these entities are identifiers, functions, and primitive types. By allowing the model to identify such entities, the model can recognize where in the structure of the source code each entity should be. For instance, it recognizes that a type identifier precedes a function entity in C++. This would essentially improve monolingual encoders. However, we do not believe it can fully replace self-supervised training objectives such as MLM. Instead, it can be considered a fine-tuning step on top of MLM to improve the model's embedding capabilities further.
>
>
> **Back translation cheat avoidance**
>
> Back Translation (BT) technique has been used in unsupervised translation-related works both for natural language and code translation. We couple this training objective with the denoising auto-encoding (DAE) training objective in a way that the model is not solely trained on one of these objectives. Instead, during training, the model iterates between DAE and BT for each batch of data. This, in hindsight, prevents the model from solely depending on BT and cheating. To further investigate this, we looked into intermediate outputs in back translation (Language B, as you mentioned).
> For example, this is one of the C++ input codes to the model (language A):
> ```
> static void makexgraph(graph *g, xword *h, int n) {
>     setword gi;
>     int i, j;
>     xword hi;
>
>     for (i = 0; i < n; ++i) {
>         hi = 0;
>         gi = g[i];
>         while (gi) {
>             j = FIRSTBITNZ(gi);
>             gi ^= bit[j];
>             hi |= XBIT(j);
>         }
>         h[i] = hi;
>     }
> }
> ```
>
> And this is the intermediate result (language B):
>
> ```
> __global__ void makexgraph(graph *g, xword *h, int n) {
>     setword gi;
>     int i = blockIdx.x * blockDim.x + threadIdx.x;
>     xword hi;
>
>     for (; i < n; i += blockDim.x * gridDim.x) {
>         hi = 0;
>         gi = g[i];
>
>         while (gi) {
>             j = FIRSTBITNZ(gi);
>             gi ^= bit[j];
>             hi |= XBIT(j);
>         }
>
>         h[i] = hi;
>     }
> }
> ```
>
> We can see that the model has tried to translate the code to CUDA code. However, the translated code contains issues. For instance, variable `j` is not defined. In the context of back translation, this noisy translated CUDA code will be used as an input to the model, and the model is responsible for reconstructing the original input C++ code. Since we iterate over languages with back translation, sometimes the model has to generate noisy CUDA code and sometimes C++ code. This step also enables the model to be able to translate from noisy input data.
>
>
> **Details on tokenizer**
>
> BPE is one of the most potent and popular types of tokenizers. For CodeRosetta, we loaded a pre-trained BPE tokenizer from uniXcoder[1] and then trained it further on our trainsets. Training tokenizers from scratch is notoriously time-consuming; this is why we load a pre-trained tokenizer that has already seen some C++ data.
>
> [1]Guo, D., Lu, S., Duan, N., Wang, Y., Zhou, M., & Yin, J. (2022). Unixcoder: Unified cross-modal pre-training for code representation. arXiv preprint arXiv:2203.03850.

---

> > ### Comment · Reviewer_9e24 · 2024-08-12
> > **Thanks for the response**
> >
> > Thank you for performing the detailed analysis and not shying away from the limitations of BLEU. I would encourage the authors to add this discussion to the manuscript. Such incidents can add a fair amount of noise to the evaluations. However, the improvements achieved by the work are considerable and I remain cautiously optimistic for the paper.

---

> > > ### Author Response · Authors · 2024-08-12
> > > **Thank you for the support!**
> > >
> > > Dear Reviewer 9e24,
> > >
> > > Thank you for your thoughtful review and for recognizing the significance of our contribution. We appreciate your support and the time you took to engage with our detailed analysis. Your feedback has been instrumental in improving our work.
> > >
> > > We will incorporate the discussion on BLEU and the potential noise in evaluations into the final version of the manuscript, as you suggested.
> > >
> > > Thank you once again for your encouraging feedback and supporting our work.
> > >
> > > Best Regards,
> > >
> > > The Authors

---

### Author Rebuttal · Authors · 2024-08-07

Dear reviewers,

Thank you very much for your invaluable feedback and comments. We acknowledge that evaluation metrics may not capture all nuances and that systematically evaluating the generated code against references is challenging. However, these metrics have been used widely in the baseline papers. Nonetheless, to address this issue, we manually analyzed the models’ results to reveal how CodeRosetta (0.8B) differs from other proprietary models.

We analyzed 30 randomly selected CUDA kernels by creating unique template programs using reference code and verifying the outputs of each generated CUDA code. The code was executed using NVCC. Our experiments show that CodeRosetta can generate functionally correct code, and we provide insights from our manual execution of translated code. Due to limitations, we only show the code snippet rather than the full kernel.

---

**Case 1**

Reference CUDA code:

`__global__ void fill_kernel(int N, float ALPHA, float *X, int INCX) {int i = (blockIdx.x + blockIdx.y * gridDim.x) * blockDim.x + threadIdx.x;`


CodeRosetta CUDA translation:
`int i = (blockIdx.x + blockIdx.y * gridDim.x) * blockDim.x + threadIdx.x;`


GPT-4 Translation:
`int i = blockIdx.x * blockDim.x + threadIdx.x;`

Gemini Pro Translation:
`int i = blockIdx.x * blockDim.x + threadIdx.x;`

Gemini Ultra Translation:
`int i = blockIdx.x * blockDim.x + threadIdx.x;`

This kernel is designed to be launched with a grid of thread blocks. Each thread calculates its global index `i`, and if `i` is within the array's bounds (`i < N`), it assigns the value `ALPHA` to the element at index `i * INCX` in the array `X`. The translated code from CodeRosetta correctly identifies the 2D grid structure with `(blockIdx.x + blockIdx.y * gridDim.x) * blockDim.x + threadIdx.x`, while other models use a 1D structure with `blockIdx.x * blockDim.x + threadIdx.x`. The choice of grid structure significantly impacts CUDA performance, and our model successfully finds the optimized result similar to the baseline. Unlike the other models, we observed four other instances where CodeRosetta used the correct grid structure.

---

**Case 2**

Reference CUDA code :
`__global__ void set_sorting_offset(const int nrows, const int ncols, int *offsets) {int tid = threadIdx.x + blockIdx.x * blockDim.x;`

CodeRosetta Translation:
`int tid = blockIdx.x * blockDim.x + threadIdx.x;`

Gemini Ultra Translation:
`int tid = threadIdx.x;`

The purpose of this kernel is to initialize an array of offsets for sorting, where each offset corresponds to the starting position of a column in a flattened 2D grid. This is useful for parallel sorting or column-wise operations. Using `threadIdx.x + blockIdx.x * blockDim.x;` gives each thread a unique index across the entire grid, suitable for accessing unique elements in a global array. In contrast, `threadIdx.x;` only provides a block-unique index, risking data races when accessing global data. The code translated from Gemini Ultra suffers from this issue, causing data races and failing to fulfill the kernel's intention. It behaves in a similar fashion over some other kernels too. Other models have successfully translated the code correctly.

---

**Case 3**

Reference CUDA code:

`__global__ void opL23 ( float * vec , float * vec1 , long depth , long rows , long cols ) { unsigned long x = threadIdx.x + blockIdx.x * blockDim.x ; unsigned long y = threadIdx.y + blockIdx.y * blockDim.y; unsigned long z`

CodeRosetta translation:
`unsigned long x = blockIdx.x * blockDim.x + threadIdx.x ;`

GPT-4 Translation:
`int x = blockIdx.x * blockDim.x + threadIdx.x;`

Gemini Pro Translation:
`int i = threadIdx.x + blockIdx.x * blockDim.x;`

Gemini Ultra Translation:
`int x = blockIdx.x * blockDim.x + threadIdx.x;`

This kernel function processes 3D arrays in parallel. Each thread calculates its 3D position, performs bounds checks, and updates specific elements of the `vec` array based on `vec1`. It averages and scales values from `vec1` and stores the results in `vec`, ensuring safe memory access within array limits. CodeRosetta ensures optimal translation without index overflow by using `unsigned long`, unlike GPT-4 and Gemini Ultra, which fail with large block and grid dimensions due to `int` usage. Gemini Pro absolutely fails in this translation producing erroneous results. Gemini Pro also misses the `const` type qualifier over different kernels, overwriting read-only data.

---

**Case 4**

We also analyze some examples from Fortran to C++ translation in the test set.
Reference C++ Code:

`#include <stdio.h>
#include <omp.h>
int main() {
    int x = 0, y;
    #pragma omp parallel`

CodeRosetta Translation:
`#include <omp.h>\n\nint main(){#pragma omp parallel`

Gemini Pro:
`#include <omp.h>\nint main(){#pragma omp parallel`

Gemini Ultra:
`#include <omp.h>\n\nint main() {#pragma omp parallel`

GPT4 Generated Translation:

`#include <atomic>
#include <thread>
#include <mutex>
std::mutex mtx;`

The code snippets are similar in functionality, showing synchronization of shared variables between threads. The main differences are the synchronization primitives used. OpenMP uses directives (`#pragma omp critical`, `#pragma omp flush`, `#pragma omp atomic`) for synchronization and memory visibility. C++ uses `std::mutex, std::atomic`, and `std::atomic_thread_fence` to achieve the same objectives from their headers.

Both approaches ensure `x` is correctly updated and visible to the second thread before it prints its value, synchronizing the threads' actions. The Fortran code included OMP, which CodeRosetta, Gemini Pro, and Gemini Ultra identified, but GPT-4 did not, instead using a different method. This highlights the limitations of BLEU like metrics, which focus on syntax rather than functionality. Despite the functional equivalence, GPT-4's code would score lower. This underscores the need for human evaluation to ensure code correctness, as no automated metric or benchmark can fully capture this.

---

### Decision · Program_Chairs · 2024-09-25

**Decision:**

Accept (poster)

**Comment:**

The paper investigates the problem of unsupervised code translation, focusing on parallel programming languages like C++ and CUDA. By introducing novel pre-training objectives like AST Entity Recognition and customized Denoising Auto-Encoding, CodeRosetta effectively learns parallel programming syntax. Experimental results demonstrate that CodeRosetta outperforms state-of-the-art methods in C++ to CUDA translation with relatively small models (0.8B). The reviewers generally appreciate the method's performance and the newly introduced training objectives. Reviewers' concerns were mostly addressed during the rebuttal.